# MASSIVE-STEPS: MASSIVE SEMANTIC TRAJECTORIES FOR UNDERSTANDING POI CHECK-INS

## ABSTRACT

Understanding human mobility through Point-of-Interest (POI) trajectory modeling is increasingly important for applications such as urban planning, personalized services, and generative agent simulation. However, progress in this field is hindered by two key challenges: the over-reliance on older datasets from 2012-2013 and the lack of reproducible, city-level check-in datasets that reflect diverse global regions. To address these gaps, we present Massive-STEPS (Massive Semantic Trajectories for Understanding POI Check-ins), a large-scale, publicly available benchmark dataset built upon the Semantic Trails dataset and enriched with semantic POI metadata. Massive-STEPS spans 15 geographically and culturally diverse cities and features more recent (2017-2018) and longer-duration (24 months) check-in data than prior datasets. We benchmarked a wide range of POI models on Massive-STEPS using both supervised and zero-shot approaches, and evaluated their performance across multiple urban contexts. By releasing Massive-STEPS, we aim to facilitate reproducible and equitable research in human mobility and POI trajectory modeling. Our code is available at: `https://anonymous.4open.science/r/Massive-STEPS/`.

## 1 INTRODUCTION

**Importance of Human Mobility Data and Modeling**    Human mobility data and POI trajectory modeling are essential for understanding how individuals interact with and move through physical spaces. This understanding enables a wide range of applications, including urban planning (Yuan et al., 2025), travel service recommendations (Feng et al., 2025), improved commercial advertising strategies (Yang et al., 2022b), and Point-of-Interest (POI) recommendation (Ding et al., 2020; Li et al., 2024; Zhang et al., 2025). Recently, human mobility data has become even more crucial with the increasing use of large language model (LLM) agents to simulate human-like behavior and routines (Zhou et al., 2024; Jiawei et al., 2024). However, while simulated and aggregated human mobility data are starting to gain popularity (Feng et al., 2020; Qin et al., 2023; Stanford et al., 2024; Jiang et al., 2025), they may not accurately reflect real-world, fine-grained individual human behavior (Salim et al., 2020), highlighting the value of evaluating on real-world data. These advancements are enabled by and large with Location-based Social Networks (LBSNs), which generate vast amounts of spatio-temporal data through user check-ins (Zhang et al., 2025; Li et al., 2024). This rich data source has allowed the development of POI recommendation systems that leverage users' historical visiting behaviors to suggest relevant locations. Such systems enhance user engagement through personalization and provide commercial value to both users and businesses by aligning recommendations with individual preferences and available services (Ding et al., 2020).

**Literature Gaps**    Our paper addresses three critical gaps in POI trajectory modeling research and datasets. First, as shown in Fig. 4, the field remains dominated by studies focused on just two cities, New York and Tokyo, based on the Foursquare dataset curated by Yang et al. (2014). This dataset, collected in 2012-2013, raises concerns about its temporal quality, as many POIs may no longer exist and user behavior may have changed (Yeow et al., 2021). While some recent studies have expanded to other cities (Zhang et al., 2024a; Merinov & Ricci, 2024; Feng et al., 2025), they often rely on the Global-scale Check-in Dataset (GSCD) (Yang et al., 2015; 2016), which, despite its large coverage, is also from 2012-2013 and contains nearly 50% erroneous entries (Monti et al., 2018).

Figure 1: **Massive-STEPS Benchmark Tasks.**

Second, most existing studies are difficult to reproduce, either due to the lack of clearly defined geographic boundaries or the unavailability of the datasets themselves, hindering fair comparison and replication. Finally, we join recent efforts (Yuan et al., 2025) in advocating for the inclusion of low-resource and underrepresented cities. Expanding beyond well-studied urban centers is essential for building more generalizable and universally applicable POI models. Table 1 summarizes these limitations in terms of geographic coverage, temporal span, and reproducibility.

**Massive-STEPS Dataset**   In this paper, we introduced the Massive Semantic Trajectories for Understanding POI Check-ins (Massive-STEPS) Dataset, derived from the Semantic Trails dataset (STD) (Monti et al., 2018). Massive-STEPS includes high-quality check-ins from 2012-2013 and 2017-2018, providing more modern and updated POI check-in data. This supports longitudinal POI trajectory modeling studies and addresses the limitations of older datasets commonly used in prior studies. The dataset covers 15 diverse cities across multiple regions, including East, West, and Southeast Asia, North and South America, Australia, the Middle East, and Europe. Notably, we placed a deliberate emphasis on under-explored regions by including cities such as Jakarta, Kuwait City, and Petaling Jaya, filling a key gap in POI trajectory research that has largely focused on major urban centers. We further enriched STD by aligning it with Foursquare's Open Source Places dataset, incorporating missing metadata such as POI coordinates, POI names, and addresses.

**Benchmark Tasks**   To demonstrate the utility of this dataset, we conducted an extensive benchmark on three tasks: (1) supervised POI recommendation, (2) zero-shot POI recommendation, and (3) spatiotemporal classification and reasoning. Our benchmark covers a wide range of models, including traditional approaches, deep learning-based models, and more recent LLM-based methods. The goal of POI recommendation task is to predict a set of POIs that a user is likely to visit based on their current check-in trajectory and historical behavior. This reflects real-world applications such as personalized POI recommendations in location-based services. Similarly, the goal of spatiotemporal classification and reasoning is to assess how effectively models (e.g., LLMs) leverage, interpret, and reason about POI trajectories. In addition, the scale of our dataset allows us to examine how urban features influence POI modeling accuracy. Building on prior hypotheses, we propose a new insight: cities with more evenly distributed POI categories tend to be harder to model, as the absence of a dominant POI category makes user behavior less predictable.

**Contribution**   This paper introduces the Massive Semantic Trajectories for Understanding POI Check-ins (Massive-STEPS) dataset, addressing gaps in existing POI trajectory modeling research. Current POI check-in datasets are often only from 2012-2013, skewed to a few cities, and lack semantic metadata, hindering the development of robust and globally applicable models. While datasets like GSCD and STD offer broad geographic coverage, they either suffer from an older timespan, contain erroneous data, or have missing information. Massive-STEPS overcomes these issues by providing high-quality check-ins from 2012-2013 and 2017-2018, improving temporal quality for longitudinal POI trajectory modeling studies. The dataset spans 15 diverse cities across multiple regions, with a focus on low-resource cities overlooked in previous research. Additionally, Massive-STEPS is enriched with metadata through alignment with Foursquare's Open Source Places, providing crucial details such as POI geographical coordinates, POI names, and addresses. We also conducted an extensive benchmark on both supervised and zero-shot POI recommendation and trajectory classification tasks, evaluating a wide range of models, including traditional methods, deep

Table 1: **Comparison of check-in datasets commonly used for POI modeling tasks**. [‡]GSCD (Yang et al., 2014; 2016) and Semantic Trails (Monti et al., 2018) are global datasets not grouped into individual cities, whereas others perform city-level grouping. [†]Replicable indicates whether city boundaries are clearly defined or can be reliably reconstructed.

| Dataset | Scale | | | Completeness | Usability | |
|---|---|---|---|---|---|---|
| | #cities | Years | #months | POI Attributes | Replicable[†] | Open-source |
| GSCD (Yang et al., 2014; 2016) | Varies[‡] | 2012-2013 | 17 | Coordinates, Category | N/A | ✓ |
| Semantic Trails (Monti et al., 2018) | Varies[‡] | **2012-2013, 2017-2018** | **24** | Category | N/A | ✓ |
| NYC and Tokyo (Yang et al., 2014) | 2 | 2012-2013 | 11 | Coordinates, Category | ✓ | ✓ |
| Gowalla-CA (Yuan et al., 2013) | 1 | 2009-2010 | 21 | Coordinates, Category | ✓ | ✓ |
| AgentMove (Feng et al., 2025) | 12 | 2012-2013 | 17 | Coordinates, Category | ✗ | ✗ |
| **Massive-STEPS** | **15** | **2012-2013, 2017-2018** | **24** | **Coordinates, Category, Name, Address** | ✓ | ✓ |

learning approaches, and recent LLM-based techniques. We further analyzed which urban features affect POI modeling accuracy and found that cities with no dominant POI category tend to be harder to predict. By releasing this dataset and benchmark code publicly, we facilitate open and reproducible research, enabling future advancements in POI trajectory modeling studies.

## 2 RELATED WORKS

### 2.1 EXISTING DATASETS

A survey conducted by Zhang et al. (2025) outlines the landscape of POI trajectory modeling research, covering a wide range of models and architectures used in prior studies. While it offers a high-level overview of the datasets used, it lacks a dedicated discussion or evaluation of POI datasets. We address this gap by analyzing commonly used datasets and positioning our dataset within this context.

**LBSN Check-in Data Sources**  Building on the tabular summary provided by Zhang et al. (2025), which offers a representative overview of the broader literature, we investigated which datasets are most commonly used in prior studies. From their original table (Table IV), we filtered entries pertaining specifically to POI and next POI recommendation tasks and identified (1) the most frequently used LBSN check-in data sources and (2) the most commonly studied cities. As shown in Fig. 4, Foursquare remains the dominant source of LBSN data in existing studies, appearing in almost 50% of the surveyed works. While several variants of Foursquare datasets have been employed, the most widely used are the NYC and Tokyo Dataset (Yang et al., 2014) (often abbreviated as FSQ-NYC and FSQ-TKY) and the Global-scale Check-in Dataset (GSCD) (Yang et al., 2015; 2016), curated by the same authors. Other LBSN sources occasionally used include Gowalla (Cho et al., 2011), Brightkite (Cho et al., 2011), and Weeplaces (Liu et al., 2017).

**Saturated to Two Cities and Old Timespan**  Due to the widespread use of FSQ-NYC and FSQ-TKY (Yang et al., 2014), the majority of POI trajectory studies are disproportionately focused on these two cities, as illustrated in Fig. 4. While there is nothing inherently problematic about NYC and Tokyo, there has been growing interest in expanding research to a broader range of cities, particularly those that are underexplored or considered low-resource (Yuan et al., 2025), as cultural and regional differences influence collective mobility behaviors. For instance, in some cities, residents tend to commute to business districts in the morning, whereas in others, nightlife activities such as visiting bars after work are more common (Yang et al., 2015). Ensuring diverse geographic coverage is increasingly important, especially as LLMs are adopted for POI trajectory modeling tasks. LLMs are known to exhibit geographical biases against regions with lower socioeconomic conditions (Manvi et al., 2024). Whether LLMs can generalize across diverse urban environments is to be investigated.

In addition, because many studies rely on the FSQ-NYC and FSQ-TKY, they are often constrained to the timespan it covers: check-in data from 2012 to 2013. However, POI data is inherently dynamic: venues may have closed, relocated, or changed in category over time. Yeow et al. (2021) underscores the importance of validating the temporal quality of POI datasets by recording whether and when a venue's information has been updated to reflect real-world changes. This is particularly critical, as

Table 2: **Statistics** of the 15 Massive-STEPS subsets, including the number of users, trajectories, POIs, check-ins, and train, validation, and test sample counts. $\mu_{\text{TrajLen}}$ denotes the mean number of check-ins per trajectory, and $\mu_{\text{interval}}$ denotes the mean time interval between check-ins (in hours). For comparison, we also include statistics from existing Foursquare- and Gowalla-based datasets. [†]Due to variations in dataset preprocessing across studies, we report the version used by Yan et al. (2023).

| City | Users | Trajectories | POIs | Check-ins | #train | #val | #test | $\mu_{\text{TrajLen}}$ | $\mu_{\text{interval}}$ |
|---|---|---|---|---|---|---|---|---|---|
| **NYC and Tokyo Check-in Dataset[†] (Yang et al., 2014)** | | | | | | | | | |
| New York | 1,048 | 14,130 | 4,981 | 103,941 | 72,206 | 1,400 | 1,347 | 7.55 | 7.27 |
| Tokyo | 2,282 | 65,499 | 7,833 | 405,000 | 274,597 | 6,868 | 7,038 | 6.32 | 5.47 |
| **Gowalla[†] (Cho et al., 2011; Yuan et al., 2013)** | | | | | | | | | |
| California | 3,957 | 45,123 | 9,690 | 238,369 | 154,253 | 3,529 | 2,780 | 5.24 | 8.37 |
| **Massive-STEPS** | | | | | | | | | |
| Bandung | 3,377 | 55,333 | 29,026 | 161,284 | 113,058 | 16,018 | 32,208 | 2.91 | 3.17 |
| Beijing | 56 | 573 | 1,127 | 1,470 | 400 | 58 | 115 | 2.57 | 3.10 |
| Istanbul | 23,700 | 216,411 | 53,812 | 544,471 | 151,487 | 21,641 | 43,283 | 2.52 | 4.36 |
| Jakarta | 8,336 | 137,396 | 76,116 | 412,100 | 96,176 | 13,740 | 27,480 | 3.00 | 2.81 |
| Kuwait City | 9,628 | 91,658 | 17,180 | 232,706 | 64,160 | 9,166 | 18,332 | 2.54 | 5.31 |
| Melbourne | 646 | 7,864 | 7,699 | 22,050 | 5,504 | 787 | 1,573 | 2.80 | 3.27 |
| Moscow | 3,993 | 39,485 | 17,822 | 105,620 | 27,639 | 3,949 | 7,897 | 2.67 | 3.36 |
| New York | 6,929 | 92,041 | 49,218 | 272,368 | 64,428 | 9,204 | 18,409 | 2.96 | 3.16 |
| Palembang | 267 | 4,699 | 4,343 | 14,467 | 10,132 | 1,487 | 2,848 | 3.08 | 3.17 |
| Petaling Jaya | 14,308 | 180,410 | 60,158 | 506,430 | 126,287 | 18,041 | 36,082 | 2.81 | 2.96 |
| São Paulo | 5,822 | 89,689 | 38,377 | 256,824 | 62,782 | 8,969 | 17,938 | 2.86 | 3.54 |
| Shanghai | 296 | 3,636 | 4,462 | 10,491 | 2,544 | 364 | 728 | 2.89 | 3.02 |
| Sydney | 740 | 10,148 | 8,986 | 29,900 | 7,103 | 1,015 | 2,030 | 2.95 | 3.33 |
| Tangerang | 1,437 | 15,984 | 12,956 | 45,521 | 32,085 | 4,499 | 8,937 | 2.85 | 3.24 |
| Tokyo | 764 | 5,482 | 4,725 | 13,839 | 3,836 | 549 | 1,097 | 2.52 | 5.16 |

recommender systems should avoid suggesting POIs that no longer exist or have undergone substantial changes (e.g., a former bookstore converted into a coworking space) and behave dynamically over longitudinal periods (Yabe et al., 2024). Moreover, behavioral patterns captured over a decade ago may no longer align with modern user preferences and routines. For example, the opening of a new train station may significantly shift commuting patterns and the popularity of surrounding POIs.

**Low Data Quality: Erroneous Entries**   More recently, researchers have begun leveraging the broader Global-scale Check-in Dataset (GSCD) (Yang et al., 2015; 2016), which spans 415 cities across 77 countries. Despite its wider geographic coverage, GSCD is temporally limited to the same 2012-2013 period as FSQ-NYC and FSQ-TKY, and thus suffers from similar issues of temporal quality. More critically, Monti et al. (2018) demonstrated that GSCD suffers from significant data quality issues, with over 14 million check-ins (about 44%) of the dataset flagged as erroneous due to anomalous user behavior. These include (1) repeated check-ins at the same venue, (2) check-ins occurring within implausibly short time intervals (less than one minute), and (3) transitions between venues that would require travel speeds exceeding Mach 1, which are physically unreasonable.

To address these limitations, Monti et al. (2018) introduced the Semantic Trails Dataset (STD), which applies systematic filtering procedures to enhance data quality. STD comprises two subsets: a cleaned version of GSCD covering 2012-2013 (STD 2013), and a newer collection of check-ins from 2017-2018 (STD 2018), sourced from Foursquare Swarm. STD 2018 also spans a wider range of cities, making it valuable for capturing globally distributed user behavior, in contrast to GSCD's focus on densely populated urban centers. Both subsets follow the same rigorous filtering criteria, resulting in a higher-quality check-in dataset for downstream POI trajectory modeling tasks. Given these improvements, we adopted STD as the source for our check-in dataset.

**Poor Reproducibility**   Another persistent challenge in POI trajectory research is the lack of reproducibility in dataset preprocessing. While some recent studies utilize datasets like GSCD to cover a wide range of cities, they often omit important details needed for replicating their data filtering processes. For example, Feng et al. (2025) and Zuo & Zhang (2024) conducted city-level filtering, but they did not specify how the city boundaries were defined or what distance-based thresholds were used. Similarly, the Weeplaces dataset used by Chen & Zhu (2025) and Cao et al. (2023) is no longer available. To further support this claim, we provide an extensive list of dataset reproducibility issues in all the studies reviewed by Zhang et al. (2025), in Table 7. As shown, **almost none** of the datasets

used in these works are fully reproducible or publicly available, except for FSQ-NYC/TKY (Yang et al., 2014) and Gowalla-CA (Yuan et al., 2013), leading to a heavy reliance on these datasets.

## 2.2 UNDERSTANDING URBAN FEATURES AND POI TRAJECTORY MODELING

POI trajectory studies that evaluate models across multiple city-level datasets often include analyses to assess how well their methods generalize across different urban contexts. It is well understood that POI recommendation accuracy metrics (e.g., Acc@k, NDCG@k) can vary substantially between cities and can be interpreted as a proxy for how easy or difficult a city is to model. The assumption is that higher performance reflects more predictable or structured mobility patterns. This viewpoint is consistent with prior work highlighting the role of cultural and urban-specific factors in shaping mobility behaviors (Yang et al., 2015; Sun et al., 2024).

Several studies have proposed hypotheses connecting specific urban features to modeling difficulty. Yang et al. (2022c) hypothesized that cities with fewer check-ins and higher spatial sparsity of POIs are harder to model. Yan et al. (2023) suggested that a larger number of user trajectories improves predictive accuracy by providing richer collaborative signals, whose architecture is designed to leverage. Li et al. (2024) proposed that cities with a greater variety of POI categories are easier to model due to LLMs' contextual reasoning capabilities, whereas cities covering a broader geographic area tend to be more difficult to model. In the zero-shot POI recommendation setting, Feng et al. (2025) reported two key findings: (1) geospatial biases inherent in LLMs can hinder prediction quality across cities, and (2) LLMs are influenced by city-specific mobility patterns.

Building on these insights, we used Massive-STEPS to explore how urban features affect POI recommendations. Its diverse set of 15 cities allows for a comprehensive analysis across different cultural and urban contexts. We analyzed the correlation between urban features and model accuracy, and based on the results, proposed a new hypothesis that contrasts previous findings in the literature.

## 3 MASSIVE-STEPS DATASET

### 3.1 CREATION PROCESS

Massive-STEPS is derived from STD (Monti et al., 2018), incorporating check-ins from both the 2013 and 2018 subsets. We utilize two additional components from STD: (1) the **cities** metadata file, which provides the latitude and longitude of administrative regions (e.g., towns, suburbs) along with their corresponding country codes obtained from GeoNames; and (2) the POI **category** mapping, which links each Foursquare Category ID to its descriptive name (e.g., "Restaurant"). Based on this metadata, each POI is thus associated with several attributes: Foursquare Place ID, Foursquare Category ID, category name, latitude/longitude of the administrative region, the administrative region name, and the country code. For anonymization purposes and model training compatibility, we apply ordinal encoding to the Place IDs and Category IDs, assigning each a unique integer index.

**Trajectory Grouping** Most POI trajectory models operate on sequences of check-ins, commonly referred to as trajectories. The model is tasked with predicting the next POIs a user is likely to visit, given the current trajectory. STD conveniently provides pre-grouped trajectories (trails) by applying a time interval-based grouping: for each user, check-ins that occur within a time interval of $\delta_\tau = 8$ hours are grouped into the same trajectory.

**Matching Trajectories to Target Cities** To obtain city-specific datasets, we matched trajectories to the target cities. For each city, we obtain geographic boundaries from OpenStreetMap and retrieve its GeoJSON file via the Overpass API. The GeoJSON file contains a polygon defining the city's boundary in latitude and longitude. Using this boundary, we filter check-ins by comparing the latitude/longitude of each POI's administrative region and retain only those that are within the city's polygon. This ensures that all retained trajectories are spatially grounded within the designated city.

**Filtering Short Trajectories and Inactive Users** To ensure data quality, we apply an additional filtering step by removing trajectories with fewer than two check-ins and excluding users with fewer than three trajectories. This prevents the model from learning from overly sparse or irrelevant data.

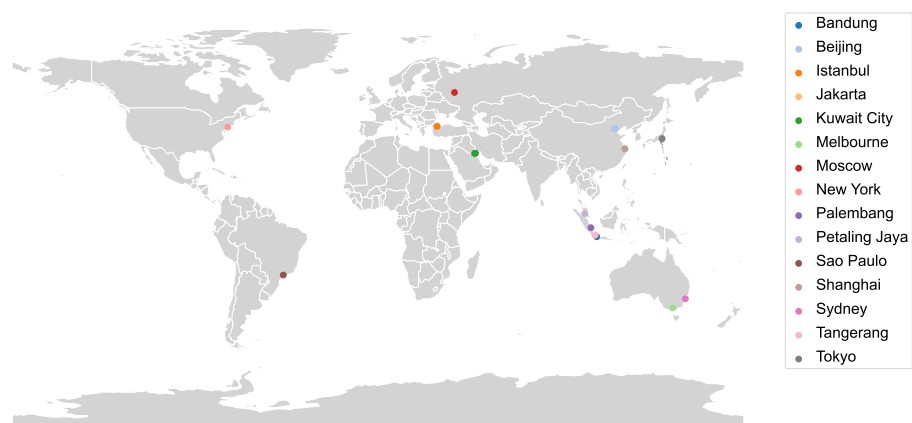

Figure 2: **World map highlighting the cities included in the Massive-STEPS dataset**.

**Train, Validation, and Test Splits**    We split trajectories into training, validation, and test sets in a ratio of 7:1:2, following Feng et al. (2025). We ensure that all users in the test set appear at least once in the training or validation set, following prior studies (Yang et al., 2022c; Yan et al., 2023).

### 3.1.1 POI ENRICHMENT VIA FOURSQUARE OS PLACES

Since the POIs in STD include their corresponding Foursquare Place IDs, we matched them directly with entries in the Foursquare OS Places dataset using these IDs as the key. This one-to-one ID correspondence allows for a straightforward join operation, enriching each POI with additional metadata such as its precise latitude and longitude, name (e.g., of a restaurant or subway station), and address. However, not all POIs in the Foursquare OS Places dataset include the full metadata, particularly those categorized as private residences, which are excluded due to privacy restrictions.

### 3.2 DESCRIPTION AND ADDRESSING LITERATURE GAPS

#### 3.2.1 PREPROCESSING

**Massive-STEPS is a city-level POI check-in dataset** comprising user check-in trajectories from 15 cities: Bandung, Beijing, Istanbul, Jakarta, Kuwait City, Melbourne, Moscow, New York, Palembang, Petaling Jaya, São Paulo, Shanghai, Sydney, Tangerang, and Tokyo. It features anonymized POI check-ins enriched with geographical metadata to support spatiotemporal and sequential modeling tasks. City-level statistics, along with comparisons to existing datasets, are presented in Table 2. Fig. 2 shows a world map highlighting the locations of all cities included in the dataset. Table 8 shows the available fields in the dataset and provides an example for each field.

**Massive-STEPS offers a more comprehensive and diverse representation of urban mobility** compared to typical POI check-in datasets. As shown in Table 2, datasets like FSQ-NYC and FSQ-TKY (Yang et al., 2014) contain fewer than 10,000 candidate POI locations. In contrast, cities in Massive-STEPS cover significantly more POIs: Massive-STEPS New York has over 49,000 POIs, while Massive-STEPS Jakarta exceeds 76,000. Massive-STEPS Istanbul, one of the largest subsets, features a large user base of 23,700, offering a broad range of user behaviors. Although some Massive-STEPS subsets are smaller than their FSQ counterparts (e.g., Tokyo), we attribute this to the strict filtering procedures applied by STD to remove erroneous entries, as explained in Section 2.1. This scale introduces additional computational challenges. For instance, models that rely on dense POI-to-POI adjacency matrices require efficient implementations to reduce memory consumption.

Another key feature of Massive-STEPS is its temporal coverage, covering the periods 2012-2013 and 2017-2018 (24 months in total). This enables longitudinal analyses, such as evaluating how POI models perform across different time periods (see Section 4.2). POIs are highly dynamic, with substantial closure rates in major cities like New York, Melbourne, and Sydney (Table 9), highlighting that the POI landscape is far from static and reinforcing the need for multi-period datasets like ours.

Table 3: **Benchmark results on POI recommendation task**. The metric reported is Acc@1. Full results, including other metrics, are available in Section C.4. **Bold** indicates the best performance for each city, while underline indicates the second-best.

| Model | Bandung | Beijing | Istanbul | Jakarta | KC | Melbourne | Moscow | NY | Palembang | PJ | SP | Shanghai | Sydney | Tangerang | Tokyo |
|---|---|---|---|---|---|---|---|---|---|---|---|---|---|---|---|
| FPMC | 0.048 | 0.000 | 0.026 | 0.029 | 0.021 | 0.062 | 0.059 | 0.032 | 0.102 | 0.026 | 0.030 | 0.084 | 0.075 | 0.104 | 0.176 |
| RNN | 0.062 | 0.085 | 0.077 | 0.049 | 0.087 | 0.059 | 0.075 | 0.061 | 0.049 | 0.064 | 0.097 | 0.055 | 0.080 | 0.087 | 0.133 |
| LSTPM | 0.110 | 0.127 | 0.142 | 0.099 | 0.180 | 0.091 | 0.151 | 0.099 | 0.114 | 0.099 | 0.158 | 0.099 | 0.141 | 0.154 | 0.225 |
| DeepMove | 0.107 | 0.106 | 0.150 | 0.103 | 0.179 | 0.083 | 0.143 | 0.097 | 0.084 | 0.112 | 0.160 | 0.085 | 0.129 | 0.145 | 0.201 |
| GETNext | 0.179 | 0.433 | 0.146 | 0.155 | 0.175 | 0.100 | 0.175 | 0.134 | 0.158 | 0.139 | 0.202 | 0.115 | 0.181 | 0.224 | 0.180 |
| STHGCN | 0.219 | 0.453 | 0.241 | 0.197 | 0.225 | 0.168 | 0.223 | 0.146 | 0.246 | 0.174 | 0.250 | 0.193 | 0.227 | 0.293 | 0.250 |
| UniMove | 0.007 | 0.036 | 0.015 | 0.004 | 0.023 | 0.008 | 0.009 | 0.004 | 0.009 | 0.008 | 0.002 | 0.000 | 0.015 | 0.001 | 0.032 |

We also observe distributional shifts in the most visited POI categories between the two periods in Fig. 6, indicating that visitation behaviors can change substantially even within the same city.

Beyond scale, Massive-STEPS addresses the oversaturation of FSQ-NYC and FSQ-TKY in POI trajectory modeling research. Notably, Massive-STEPS includes low-resource and previously underexplored cities in human mobility studies, such as Petaling Jaya and Kuwait City, both of which are among the cities with the highest number of check-ins from STD. This broader coverage opens new research opportunities for studying location-based behaviors across diverse cultural and geographic contexts. Furthermore, since Massive-STEPS is based on STD, it benefits from the carefully filtered, high-quality check-ins and a longer, more recent timespan. These characteristics make Massive-STEPS a more relevant and reliable resource for modeling human mobility patterns.

**Massive-STEPS is designed to be easily extended to other geographical regions**. Since the data processing code is open-source and fully reproducible, adding a new city only requires its geographic boundaries from OpenStreetMap. Moreover, Massive-STEPS is scalable to higher levels of geographic granularity, enabling the creation of provincial, state, and country-level POI check-in datasets, which support collective mobility studies at broader geographic scales.

# 4 BENCHMARK TASKS

## 4.1 POI RECOMMENDATION

This benchmark focuses on POI recommendation, where the goal is to predict a user's next visit based on their previous check-ins. The input is a trajectory of visited places, and the model is expected to suggest a set of $K$ POIs the user might visit next. It is a **supervised** task, trained on all available historical trajectories to learn personalized movement patterns. Appendix C provides details on problem formulation, hyperparameters, experimental setups, and full evaluation results.

**Experimental Setup**   We adopted the predefined trajectories from the original STD, where check-ins are grouped into sequences based on fixed time intervals (see Section 3.2.1). All input features are numerically encoded, enabling straightforward use across experiments. Models typically use four feature types: (1) social: user ID; (2) spatial: POI ID and geographic coordinates; (3) temporal: check-in timestamp; and (4) categorical: POI category. As not all POIs have exact geographic coordinates (see Section 3.1.1), we used the geographic coordinates of their administrative region as a proxy for all POIs. We evaluated four kinds of architectures: (1) Markov-based methods: FPMC (Rendle et al., 2010), (2) classical deep learning models: RNN (Wang et al., 2021a), LSTPM (Sun et al., 2020), and DeepMove (Feng et al., 2018), (3) Transformer-based graph neural networks: GETNext (Yang et al., 2022c) and STHGCN, and (4) Trajectory foundation model: UniMove (Han et al., 2025b). We employed two commonly used metrics in POI recommender systems: Acc@k, which checks if the true POI appears in the top-k predicted results, and NDCG@k, which measures the ranking quality of the suggested results.

**Results**   As shown in Table 3, STHGCN achieves the highest average Acc@1 across all cities, followed closely by GETNext, demonstrating the effectiveness of GNNs. The top model attained a mean Acc@1 of 23.4%, comparable to previous studies on similarly sized datasets (Feng et al., 2025). Notably, pre-training UniMove (Han et al., 2025b) from scratch struggled to surpass recurrent

Table 4: **Benchmark results on zero-shot POI recommendation task**. The metric reported is Acc@1. Full results, including other metrics, are available in Section E.4. **Bold** indicates the best performance for each city, while underline indicates the second-best.

| Method | LLM | Bandung | Beijing | Istanbul | Jakarta | KC | Melbourne | Moscow | NY | Palembang | PJ | SP | Shanghai | Sydney | Tangerang | Tokyo |
|---|---|---|---|---|---|---|---|---|---|---|---|---|---|---|---|---|
| LLM-Mob | **Gemini 2 Flash** | 0.105 | 0.115 | 0.080 | 0.100 | 0.095 | 0.060 | 0.130 | 0.095 | 0.135 | 0.090 | 0.130 | 0.055 | 0.060 | 0.155 | 0.140 |
| | **Qwen 2.5 7B** | 0.060 | 0.058 | 0.035 | 0.105 | 0.080 | 0.030 | 0.090 | 0.070 | 0.075 | 0.030 | 0.090 | 0.040 | 0.035 | 0.095 | 0.110 |
| | **Llama 3.1 8B** | 0.010 | 0.000 | 0.020 | 0.055 | 0.030 | 0.010 | 0.030 | 0.025 | 0.005 | 0.010 | 0.030 | 0.005 | 0.020 | 0.020 | 0.005 |
| | **Gemma 2 9B** | 0.070 | 0.115 | 0.075 | 0.105 | 0.080 | 0.055 | 0.100 | 0.070 | 0.095 | 0.055 | 0.085 | 0.050 | 0.030 | 0.145 | 0.145 |
| LLM-ZS | **Gemini 2 Flash** | 0.095 | 0.058 | 0.090 | 0.110 | 0.080 | 0.065 | 0.125 | 0.080 | 0.130 | 0.110 | 0.150 | 0.065 | 0.060 | 0.145 | 0.160 |
| | **Qwen 2.5 7B** | 0.055 | 0.038 | 0.040 | 0.065 | 0.050 | 0.040 | 0.080 | 0.050 | 0.050 | 0.045 | 0.095 | 0.045 | 0.045 | 0.100 | 0.120 |
| | **Llama 3.1 8B** | 0.045 | 0.077 | 0.040 | 0.045 | 0.060 | 0.040 | 0.080 | 0.055 | 0.070 | 0.030 | 0.030 | 0.060 | 0.040 | 0.080 | 0.110 |
| | **Gemma 2 9B** | 0.065 | 0.096 | 0.045 | 0.105 | 0.070 | 0.050 | 0.080 | 0.075 | 0.060 | 0.065 | 0.075 | 0.050 | 0.045 | 0.100 | 0.110 |
| LLM-Move | **Gemini 2 Flash** | **0.225** | 0.096 | **0.205** | **0.295** | **0.220** | **0.225** | 0.220 | **0.235** | **0.260** | **0.210** | **0.285** | **0.170** | **0.230** | **0.200** | **0.250** |
| | **Qwen 2.5 7B** | 0.100 | **0.192** | 0.175 | 0.115 | 0.160 | 0.110 | **0.230** | 0.120 | 0.130 | 0.135 | 0.155 | 0.095 | 0.125 | 0.175 | **0.250** |
| | **Llama 3.1 8B** | 0.030 | 0.058 | 0.015 | 0.015 | 0.010 | 0.040 | 0.005 | 0.035 | 0.010 | 0.040 | 0.045 | 0.020 | 0.055 | 0.000 | 0.030 |
| | **Gemma 2 9B** | 0.175 | 0.096 | 0.100 | 0.235 | 0.120 | 0.115 | 0.110 | 0.115 | 0.210 | 0.175 | 0.195 | 0.105 | 0.125 | 0.125 | 0.130 |

model baselines. We attribute this to the high number of cold-start trajectories (see Fig. 7), which hinder performance as next-token prediction loss struggles with extremely short input sequences. We also examined the impact of urban features on POI recommendation accuracy by computing Spearman correlations between city features and model performance. As shown in Fig. 9, we found that **category entropy**, based on Shannon entropy, shows a strong negative correlation with accuracy ($r = -0.684$). Cities with more evenly distributed POI categories tend to be harder to predict. This result aligns with prior findings on other datasets. Further details are provided in Appendix D.

## 4.2 ZERO-SHOT POI RECOMMENDATION

This benchmark focuses on zero-shot POI recommendation via LLMs, where the goal is to predict a user's next visit based on their previous check-ins (similar to its supervised counterpart) without additional model fine-tuning. The input is a user trajectory transformed into a textual prompt, and the model ranks a set of $K$ candidate POIs to identify the next likely destination. Appendix E provides details on problem formulation, prompts, experimental setups, and full evaluation results.

**Experimental Setup**  For zero-shot recommendation, trajectories are converted into textual prompts (Xue et al., 2022; Xue & Salim, 2024). We adapted the prompt templates from Feng et al. (2025), which implemented the three LLM methods evaluated in this study: LLM-Mob (Wang et al., 2023c), LLM-ZS (Beneduce et al., 2024), and LLM-Move (Feng et al., 2024a). Since LLMs can leverage contextual information, features do not need numerical encoding; we used each check-in's timestamp, POI category name, and POI ID. For a robust evaluation, we tested each method on four LLMs: one closed-source API (Gemini 2.0 Flash (Team & et al., 2024a)) and three open-source instruction-tuned models (Qwen 2.5 7B (Team, 2024), Llama 3.1 8B (Grattafiori et al., 2024), and Gemma 2 9B (Team & et al., 2024b)). We used the same metrics as in the supervised setting: Acc@k and NDCG@k.

**Results**  As shown in Table 4, LLM-Move (Feng et al., 2024a) outperformed the other two methods due to its prompt, which provides candidate POIs rather than relying solely on historical or contextual trajectories unlike LLM-Mob and LLM-ZS. Across LLMs, Gemini 2.0 Flash

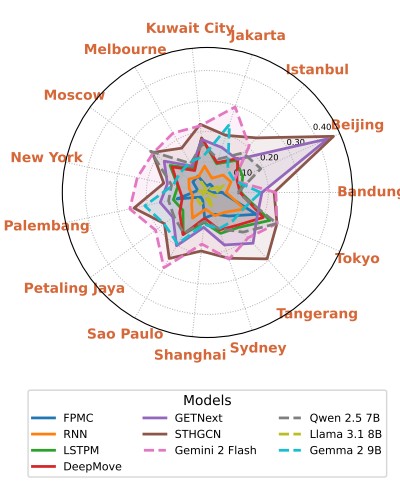

Figure 3: Acc@1 of supervised and LLM-Move models across 15 cities.

achieved the highest accuracy across all prompting strategies, with Qwen 2.5 7B and Gemma 2 9B as strong open-source alternatives. Notably, as shown in Fig. 3, these zero-shot methods matched or exceeded supervised baselines in several cities (e.g., Jakarta, Kuwait City, Moscow), demonstrating their effectiveness without fine-tuning. Although serving LLMs requires more powerful hardware, running inference can still be faster overall than training supervised models from scratch.

Table 5: **Zero-shot POI recommendation results using LLM-Move across two time periods.** The metric reported is Acc@1. Full results, including other metrics, are available in Section E.4. **Bold** indicates the best performance for each city, while underline indicates the second-best. Results marked as N/A indicate that no samples were available for that city in the corresponding time period.

| Time Period | Model | Bandung | Beijing | Istanbul | Jakarta | KC | Melbourne | Moscow | NY | Palembang | PJ | SP | Shanghai | Sydney | Tangerang | Tokyo |
|---|---|---|---|---|---|---|---|---|---|---|---|---|---|---|---|---|
| 2012-2013 | **Gemini 2 Flash** | **0.227** | 0.102 | **0.212** | **0.295** | **0.423** | **0.226** | 0.218 | **0.240** | **0.256** | **0.199** | **0.298** | **0.192** | **0.256** | **0.197** | N/A |
| | **Qwen 2.5 7B** | 0.098 | **0.204** | 0.192 | 0.114 | 0.269 | 0.116 | **0.234** | 0.130 | 0.128 | 0.142 | 0.173 | 0.109 | 0.122 | 0.172 | N/A |
| | **Llama 3.1 8B** | 0.031 | 0.041 | 0.007 | 0.010 | 0.000 | 0.039 | 0.005 | 0.032 | 0.010 | 0.014 | 0.048 | 0.006 | 0.064 | 0.000 | N/A |
| | **Gemma 2 9B** | 0.180 | 0.102 | 0.116 | 0.228 | 0.308 | 0.097 | 0.112 | 0.130 | 0.215 | **0.199** | 0.202 | 0.109 | 0.122 | 0.126 | N/A |
| 2017-2018 | **Gemini 2 Flash** | **0.167** | 0.000 | **0.185** | 0.286 | **0.190** | **0.222** | **0.333** | **0.217** | **0.400** | **0.237** | **0.219** | **0.091** | **0.136** | **0.500** | **0.250** |
| | **Qwen 2.5 7B** | **0.167** | 0.000 | 0.130 | 0.143 | 0.144 | 0.089 | 0.000 | 0.087 | 0.200 | 0.119 | 0.063 | 0.045 | **0.136** | **0.500** | **0.250** |
| | **Llama 3.1 8B** | 0.000 | **0.333** | 0.037 | 0.143 | 0.011 | 0.044 | 0.000 | 0.043 | 0.000 | 0.102 | 0.031 | 0.068 | 0.023 | 0.000 | 0.030 |
| | **Gemma 2 9B** | 0.000 | 0.000 | 0.056 | **0.429** | 0.092 | 0.178 | 0.000 | 0.065 | 0.000 | 0.119 | 0.156 | **0.091** | **0.136** | 0.000 | 0.130 |

Table 6: **Benchmark results on spatiotemporal classification task**. The metric reported is Acc. **Bold** indicates the best performance for each city, while underline indicates the second-best.

| LLM | Bandung | Beijing | Istanbul | Jakarta | KC | Melbourne | Moscow | NY | Palembang | PJ | SP | Shanghai | Sydney | Tangerang | Tokyo |
|---|---|---|---|---|---|---|---|---|---|---|---|---|---|---|---|
| **Gemini 2 Flash** | **0.635** | 0.615 | **0.715** | **0.650** | **0.765** | **0.635** | 0.740 | **0.620** | **0.670** | **0.610** | **0.730** | **0.600** | **0.550** | **0.635** | 0.510 |
| **GPT-4o Mini** | 0.625 | 0.538 | 0.610 | 0.610 | 0.430 | **0.635** | **0.745** | 0.600 | 0.645 | 0.590 | 0.645 | 0.565 | 0.545 | 0.600 | 0.495 |
| **GPT-4.1 Mini** | 0.585 | **0.673** | 0.615 | 0.600 | 0.690 | 0.585 | **0.745** | 0.595 | 0.605 | 0.575 | 0.700 | 0.565 | 0.515 | 0.620 | 0.550 |
| **GPT-5 Nano** | 0.570 | 0.635 | 0.535 | 0.530 | 0.470 | 0.500 | 0.635 | 0.580 | 0.560 | 0.565 | 0.680 | 0.465 | 0.440 | 0.520 | **0.580** |

**Longitudinal Experiments and Results** To examine temporal changes in mobility patterns, we split the test sets into two periods (2012-2013 and 2017-2018) and evaluated the same four LLMs with LLM-Move (Feng et al., 2024a), which was the strongest-performing approach in our zero-shot experiments. As shown in Table 5, zero-shot accuracy generally declined in the 2017-2018 period, except for Jakarta and Tangerang, indicating that user trajectories in later years tend to be more challenging to predict. Performance trends varied across cities, highlighting temporal differences in mobility patterns that impact downstream tasks. Across all models, Gemini 2.0 Flash consistently achieved the highest accuracy, demonstrating robust zero-shot capabilities across cities and time.

## 4.3 SPATIOTEMPORAL CLASSIFICATION AND REASONING

This benchmark assesses whether LLMs can be leveraged for spatiotemporal trajectory classification by providing them with contextual information about a user's behavior. The task evaluates the model's ability to capture variations in travel patterns across different cities, given the sequence of POI check-ins as input, and without any additional fine-tuning. Through this setup, we aim to understand how effectively LLMs can reason over spatiotemporal and behavioral cues in user trajectories. Appendix F provides details on problem formulation, prompts, and LLM parameters.

**Experimental Setup** This task involves classifying a property of a POI check-in trajectory. For this study, we chose to predict whether the final check-in occurs on a weekday or a weekend. Each trajectory is converted into a textual prompt incorporating spatial (city), temporal (check-in time-of-day), and categorical contexts (POI category). Adapting the prompt design from LLM-Mob (Wang et al., 2023c), we instructed the LLM to first reason before making a prediction. This approach allows us to evaluate both classification accuracy and the spatiotemporal reasoning capabilities of LLMs, in line with recent work on spatiotemporal reasoning using LLMs (Quan et al., 2025). Whereas prior approaches rely on models that encode trajectories (Nayak & Pandit, 2023), our method directly leverages the LLM's ability to process contextual information in natural language. We evaluated four closed-source LLM APIs: Gemini 2.0 Flash (Team & et al., 2024a), GPT-4o Mini, GPT-4.1 Mini, and GPT-5 Nano (OpenAI & et al., 2024b;a), and used Accuracy as our primary metric.

**Results** As shown in Table 6, Gemini 2 Flash achieves the highest mean accuracy of 0.643 across the 15 cities. While this performance is above random guessing, it remains far from ideal for practical spatiotemporal trajectory classification. Surprisingly, the GPT series of models, despite some being more recent than Gemini 2 Flash, generally performed worse. Notably, GPT-5 Nano obtained the lowest mean accuracy, even though it is designed for advanced reasoning tasks. Our findings align with González et al. (2008), who observed that user regularity does not differ significantly between weekdays and weekends, suggesting that mobility patterns are not strictly dictated by work schedules

but may instead reflect intrinsic human activity patterns. Overall, these results indicate that current LLMs face significant limitations in capturing spatiotemporal patterns from trajectory data alone, highlighting the need for further improvements in this area.

## 5 CONCLUSION AND LIMITATIONS

**Conclusion**   In this paper, we presented the Massive-STEPS dataset to address longstanding limitations in POI trajectory modeling research, particularly the reliance on older, geographically saturated, and non-reproducible check-in datasets. Massive-STEPS offers a large-scale, semantically enriched resource spanning 15 cities across diverse global regions and two time periods, supporting both longitudinal and cross-city analyses. The dataset includes rich semantic information such as venue name, address, category, and coordinates. We also provide benchmark results for supervised and zero-shot POI trajectory modeling methods, illustrating the dataset's utility across model types and tasks. By releasing Massive-STEPS and our evaluation pipeline publicly, we aim to advance open, reproducible, and globally inclusive research in human mobility and POI trajectory modeling systems.

**Limitations**   Firstly, Massive-STEPS is derived from the Semantic Trails dataset and thus inherits its biases and potential errors, which may propagate through downstream tasks. Additionally, the dataset is sparse in several cities, which can impact model training quality and limit cross-city generalization. Secondly, Massive-STEPS focuses solely on trajectories and POI metadata, without including user demographic or social information due to privacy considerations. This restricts its applicability for personalized or socially-aware POI recommendation tasks. Thirdly, while our benchmarking covers a wide range of models and cities to emphasize replicability and geographic breadth, we did not perform extensive hyperparameter tuning, which may affect the peak performance of the models. Finally, although Massive-STEPS does not reflect present-day mobility patterns, it was designed to provide a more recent alternative to older datasets such as FSQ-NYC/TKY and GSCD (2012-2013) and to help bridge the gap toward newer, open, and extensible POI benchmarks.

## REPRODUCIBILITY STATEMENT

Dataset and evaluation reproducibility is a central claim and contribution of our paper, especially given that the field has long been hindered by their absence. We ensure reproducibility by: (1) providing detailed descriptions and code for downloading and preprocessing the data to produce the final dataset, (2) specifying model configurations, training setups, and evaluation protocols throughout the paper (see Section C, Section E, Section F), and (3) releasing the Massive-STEPS dataset creation code along with all accompanying code to replicate our experiments.

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

## A  EXISTING POI RECOMMENDATION DATASETS

To examine the trend of the usage of POI recommendation datasets, we filtered the comprehensive survey by Zhang et al. (2025) to extract studies that explicitly mention the cities used in their experiments. The resulting distribution is summarized in Table 7, which shows a strong concentration of studies focused on New York and Tokyo. Additionally, Fig. 4 visualizes the same data, highlighting the uneven distribution of city choices across studies. We also include information on the LBSN platforms used, revealing that Foursquare remains the predominant data source in the field. These

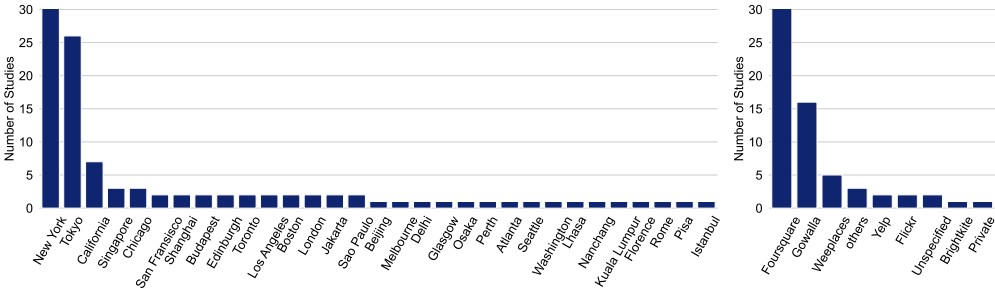

Figure 4: **Distribution of POI recommendation studies** modeled on specific cities, modified from Table IV of Zhang et al. (2025). We identified and counted studies that explicitly mentioned city names, revealing the skewness of existing research, which is saturated around New York and Tokyo. In addition, we include the distribution of studies by LBSN platform, showing that Foursquare is by far the most commonly used source of check-in data. The list of studies is shown in Table 7.

findings underscore the need for broader, more inclusive datasets that support evaluation across a wider range of global cities.

# B    DATA VISUALIZATION

We present several visualizations highlighting Massive-STEPS' scale and diversity to complement our dataset description.

In Fig. 5, we show the top 10 most frequent POI categories for each city. The distribution reflects the local culture and lifestyle across different urban areas. For example, Beijing and Shanghai have a high number of Chinese restaurants, while Melbourne and Sydney show a strong presence of cafes. In Tokyo, convenience stores and ramen shops dominate. These patterns illustrate the diversity of local culture and user interests. Fig. 6 illustrates the temporal shift in the distribution of the top 10 most visited POI categories across each city, comparing the percentage of visits to each category between the 2012-2013 and 2017-2018 periods. Table 9 shows the number of POIs ever opened according to Foursquare OS Places, the number of POIs confirmed closed by 2025, and the number of POIs that closed between 2014 and 2016, corresponding to the temporal gap in our dataset.

Fig. 7 plots the distribution of trajectory lengths (i.e., number of check-ins per trajectory). The distribution is long-tailed, with most trajectories being relatively short, similar to the original Semantic Trails dataset. This indicates that users often make only a few check-ins per outing.

Finally, we show the distribution of user activity levels, measured by the number of trajectories per user in Fig. 8. Most users exhibit cold-start behavior, contributing only a small number of trajectories. This highlights the importance of models that are robust to sparse and short user histories.

# C    POI RECOMMENDATION: TASK DETAILS

We adopt the conventional problem formulation used in prior POI recommendation studies (Zhang et al., 2025; Yang et al., 2022c; Yan et al., 2023), which defines the task as learning user preferences and routines from historical check-ins to recommend future POIs.

## C.1    PROBLEM FORMULATION

Let $\mathcal{U} = \{u_1, u_2, \ldots, u_M\}$ denote the set of users, $\mathcal{P} = \{p_1, p_2, \ldots, p_N\}$ the set of Points of Interest (POIs), and $\mathcal{T} = \{t_1, t_2, \ldots, t_K\}$ the set of timestamps, where $M, N, K \in \mathbb{N}$.

**POI Definition**    Each POI $p \in \mathcal{P}$ is represented as a tuple:

$$p = \langle \phi, \lambda, \kappa, \alpha, \beta, \gamma \rangle,$$

where:

- $\phi$ and $\lambda$ are the latitude and longitude,
- $\kappa$ is the POI category (e.g., *restaurant*, *park*),
- $\alpha$ is the unique POI identifier,
- $\beta$ is the textual address, and
- $\gamma$ is the POI name.

**Check-in Definition**    A check-in is a tuple $c = \langle u, p, t \rangle \in \mathcal{U} \times \mathcal{P} \times \mathcal{T}$, indicating that user $u$ visited POI $p$ at timestamp $t$.

**Trajectory Definition**    A trajectory for user $u$ is defined as a temporally ordered sequence of check-ins within a fixed time interval $\delta\tau = 8$ hours. Each trajectory $T_u^i(t)$ up to timestamp $t$ is defined as:

$$T_u^i(t) = \{(p_1, t_1), (p_2, t_2), \ldots, (p_k, t_k)\}$$

such that $t_1 < t_2 < \cdots < t_k = t$ and $t_k - t_{k-1} \leq \delta\tau$. Given a set of historical trajectories

$$\mathcal{T}_u = \{T_u^1, T_u^2, \ldots, T_u^L\}$$

Table 7: **Overview of POI Recommendation Studies by City and LBSN Platform.** This table is adapted from Table IV in the survey by Zhang et al. (2025) and presents a filtered list of POI recommendation studies that explicitly mention city names and their associated LBSN platforms.

| Study | Cities | LBSN | Dataset Reproducibility Issue |
|---|---|---|---|
| SSTPMF (Davtalab & Alesheikh, 2021) | New York, Tokyo | Foursquare, Gowalla | Gowalla city boundaries not reproducible. |
| ST-LSTM (Zhao et al., 2018) | California, Singapore | Brightkite, Foursquare, Gowalla | FSQ city boundaries not reproducible. Brightkite and Gowalla not grouped into cities. |
| LSMA (Wang et al., 2022) | New York, San Francisco, Tokyo | Foursquare, Weeplaces | Weeplaces no longer available. |
| DLAN (Wu et al., 2024) | New York, Tokyo | Foursquare | No issue, uses FSQ-NYC/TKY. |
| TLR-M (Halder et al., 2021) | New York, Tokyo | Foursquare | No issue, uses FSQ-NYC/TKY. |
| GETNext (Yang et al., 2022c) | New York, Tokyo, California | Foursquare, Gowalla | Only provides preprocessed NYC, missing TKY and CA. |
| CARAN (Hossain et al., 2022) | New York, Tokyo | Foursquare, Gowalla | Gowalla not grouped into cities. |
| JANICP (Zhong et al., 2022) | New York, Tokyo | Foursquare, Weeplaces | Weeplaces no longer available. |
| Li et al. (Li et al., 2022a) | New York, Tokyo | Foursquare | No issue, uses FSQ-NYC/TKY. |
| AMACF (Yang et al., 2022a) | New York, Tokyo | Foursquare, Weeplaces | Weeplaces no longer available. |
| CHA (Zang et al., 2021) | New York, Tokyo | Foursquare | No issue, uses FSQ-NYC/TKY. |
| HAT (Wu et al., 2023) | Beijing, Shanghai | Yelp, others | Used private datasets. |
| STAR-HIT (Xie & Chen, 2023) | New York | Foursquare, Gowalla | Gowalla not grouped into cities. |
| CAFPR (Halder et al., 2023) | Tokyo, California, Budapest, Melbourne | Foursquare | Uses POI themepark dataset. POI metadata (lat./lon., category) is missing. |
| TGAT (Jiang & Wu, 2023) | New York, Tokyo | Foursquare | No issue, uses FSQ-NYC/TKY. |
| MobGT (Xu et al., 2023) | New York | Foursquare, Gowalla | Used private datasets. |
| POIBERT (Ho & Lim, 2022) | Budapest, Delhi, Edinburgh, Glasgow, Osaka, Perth, Toronto | Flickr | Used private datasets. |
| AutoMTN (Qin et al., 2022) | New York, Tokyo | Foursquare | No issue, uses FSQ-NYC/TKY. |
| CCDSA (Wang et al., 2023b) | New York, Tokyo, San Francisco | Foursquare, Weeplaces | Weeplaces no longer available. |
| TDGCN (Cao et al., 2023) | Tokyo, California | Foursquare, Gowalla, Weeplaces | Weeplaces no longer available. |
| BayMAN (Xia et al., 2023) | New York | Foursquare, Gowalla | No issue, uses FSQ-NYC/TKY and Gowalla-CA. |
| ROTAN (Feng et al., 2024b) | New York, Tokyo, California | Foursquare, Gowalla | No issue, uses FSQ-NYC/TKY and Gowalla-CA. |
| TrajMoE (Han et al., 2025a) | Atlanta, Chicago, Seattle, Washington, New York, Los Angeles | Unspecified | Used private datasets. |
| UniMove (Han et al., 2025b) | Lhasa, Nanchang, Shanghai | Unspecified | Used private datasets. |
| STGCN (Han et al., 2020) | Boston, Chicago, London | Gowalla, others | Gowalla city boundaries not reproducible. Used private datasets. |
| ADQ-GNN (Wang et al., 2021b) | New York, Tokyo | Foursquare, Gowalla | Gowalla not grouped into cities. |
| HS-GAT (Zhang & Ma, 2024) | Boston, Chicago, London | Yelp, others | Yelp not grouped into cities. |
| HKGNN (Zhang et al., 2024b) | New York, Jakarta, Kuala Lumpur, São Paulo | Foursquare | FSQ city boundaries not reproducible. |
| S2GRec (Li et al., 2022b) | New York, Tokyo | Foursquare, Gowalla | Gowalla not grouped into cities. |
| GSBPL (Wang et al., 2023a) | New York, Tokyo | Foursquare, Gowalla | Gowalla not grouped into cities. |
| LSPSL (Jiang et al., 2023) | New York, Tokyo | Foursquare | No issue, uses FSQ-NYC/TKY. |
| SCL (Chen & Zhu, 2025) | Florence, Rome, Pisa, Edinburgh, Toronto | Flickr | Preprocessed Flickr dataset no longer available. |
| LLM-Move (Feng et al., 2024a) | New York, Tokyo | Foursquare | No issue, uses FSQ-NYC/TKY. |
| LLM4POI (Li et al., 2024) | New York, Tokyo, California | Foursquare, Gowalla | No issue, uses FSQ-NYC/TKY and Gowalla-CA. |
| Refine-POI (Li et al., 2025) | New York, Tokyo | Foursquare | No issue, uses FSQ-NYC/TKY. |
| GNPR-SID (Wang et al., 2025) | New York, Tokyo, California | Foursquare, Gowalla | No issue, uses FSQ-NYC/TKY and Gowalla-CA. |
| QT-Mob (Chen et al., 2025) | New York, Singapore | Foursquare, Private Telco | Only FSQ-NYC is publicly available. |
| DiffPOI (Qin et al., 2023) | Singapore, New York, Tokyo | Foursquare, Gowalla | Gowalla not grouped into cities. |
| DSDRec (Wang et al., 2024) | New York, Tokyo | Foursquare | No issue, uses FSQ-NYC/TKY. |
| Diff-DGMN (Zuo & Zhang, 2024) | Istanbul, Jakarta, São Paulo, New York, Los Angeles | Foursquare | FSQ city boundaries not reproducible. |

Table 8: **Fields available in the Massive-STEPS dataset**, including user, POI, geographic/spatial, and temporal details, along with example data for each field.

| Field | Description | Example |
|---|---|---|
| trail_id | Numeric identifier of trajectory | 2013_2866 |
| user_id | Numeric identifier of user | 90 |
| venue_id | Numeric identifier of POI venue | 185 |
| latitude | Latitude of POI venue | -33.87301862604473 |
| longitude | Longitude of POI venue | 151.20668402700997 |
| name | POI name | Sydney Town Hall |
| address | Street address of POI venue | 483 George St |
| venue_category | POI category name | City Hall |
| venue_category_id | Foursquare Category ID | 4bf58dd8d48988d129941735 |
| venue_category_id_code | Numeric identifier of POI category | 72 |
| venue_city | Administrative region name | Sydney |
| venue_city_latitude | Latitude of administrative region | -33.86785 |
| venue_city_longitude | Longitude of administrative region | 151.20732 |
| venue_country | Country code | AU |
| timestamp | Check-in timestamp | 2012-04-22 08:20:00 |

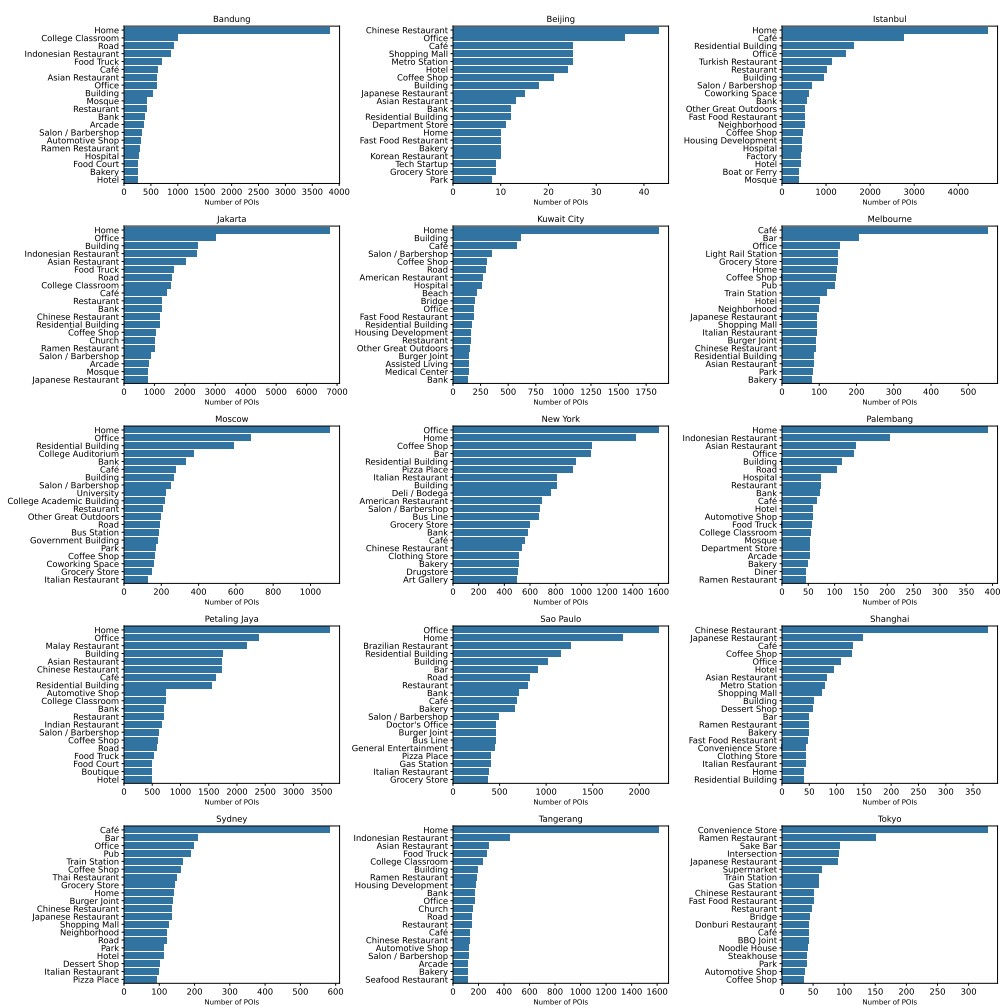

Figure 5: **Top 10 most frequent POI categories in each city**, highlighting local cultural and urban preferences.

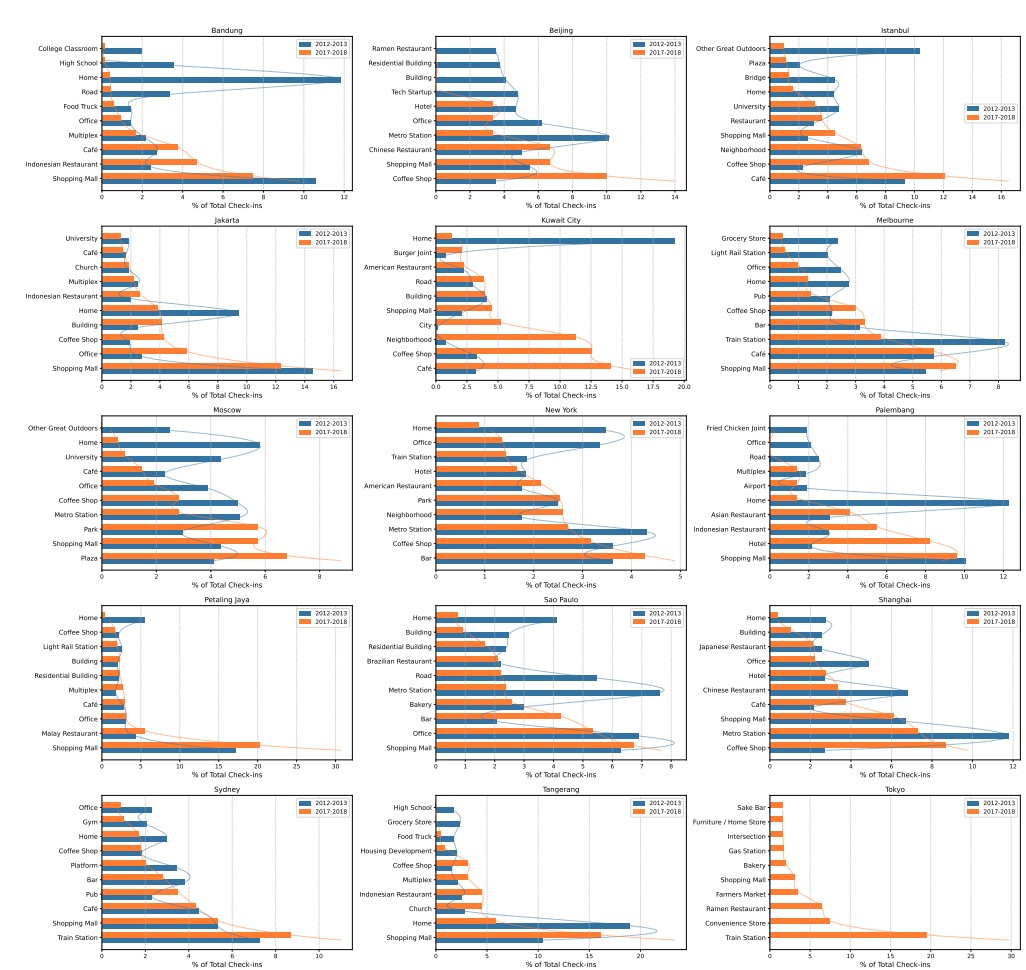

Figure 6: **Top 10 most visited POI categories in each city across two time periods**, illustrating temporal shifts in user visitation patterns.

Table 9: **Overview of POI dynamics**: total POIs ever opened (according to Foursquare OS Places), POIs confirmed closed by 2025, and POIs closed during 2014-2016, corresponding to the temporal gap in our dataset.

| City | POIs Ever Opened | Total POIs Confirmed Closed (up to 2025) | Closed within 2014-2016 |
|---|---|---|---|
| New York | 49,218 | 13,009 (26.43%) | 3,118 (6.34%) |
| Melbourne | 7,699 | 1,850 (24.03%) | 209 (2.71%) |
| Sydney | 8,986 | 1,759 (19.57%) | 253 (2.82%) |
| Moscow | 17,822 | 3,021 (16.95%) | 868 (4.87%) |
| São Paulo | 38,377 | 4,990 (13.00%) | 1,257 (3.28%) |
| Shanghai | 4,462 | 661 (14.81%) | 81 (1.82%) |
| Tokyo | 4,725 | 421 (8.91%) | 0 (0.00%) |
| Petaling Jaya | 60,158 | 4,186 (6.96%) | 1,533 (2.55%) |
| Istanbul | 53,812 | 2,833 (5.26%) | 481 (0.89%) |
| Beijing | 1,127 | 56 (4.97%) | 10 (0.89%) |
| Jakarta | 76,116 | 3,527 (4.63%) | 483 (0.63%) |
| Bandung | 29,026 | 1,053 (3.63%) | 182 (0.63%) |
| Palembang | 4,343 | 143 (3.29%) | 23 (0.53%) |
| Tangerang | 12,956 | 383 (2.96%) | 50 (0.39%) |
| Kuwait City | 17,180 | 161 (0.94%) | 22 (0.13%) |

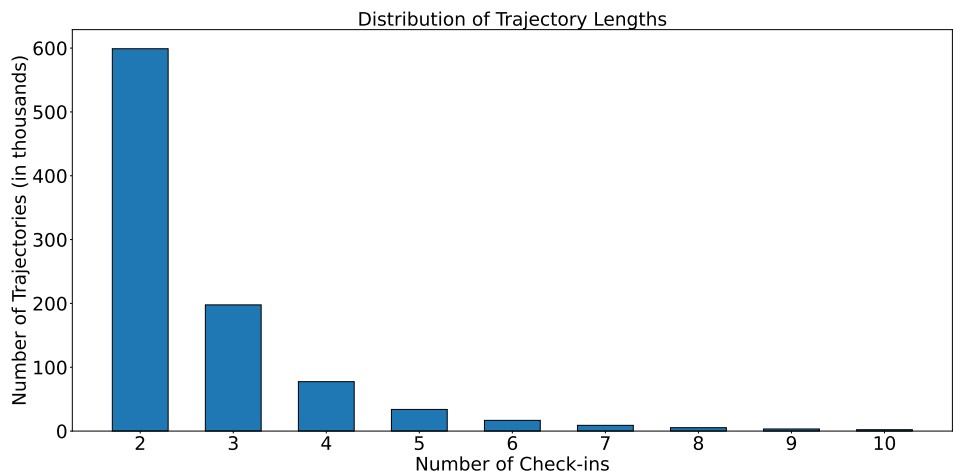

Figure 7: **Distribution of trail lengths**, showing a long-tailed pattern with most trajectories consisting of a few check-ins.

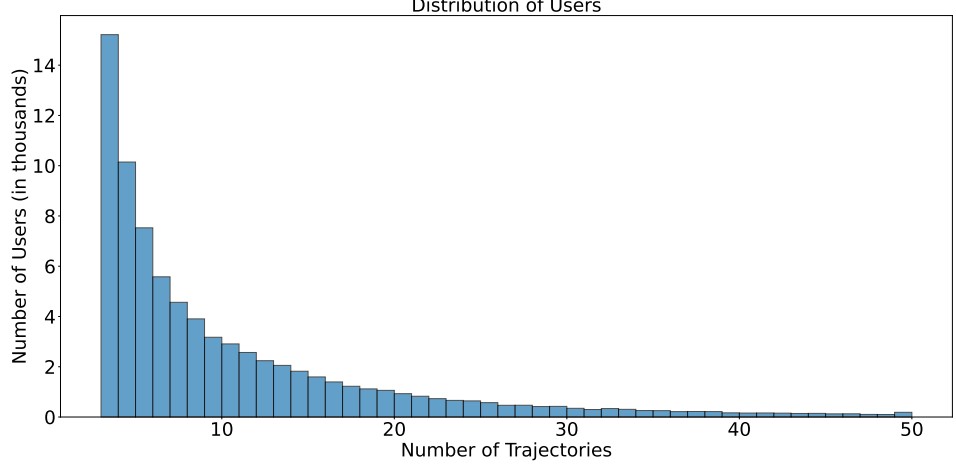

Figure 8: **Distribution of user activity** based on the number of trajectories per user, indicating a cold-start-heavy dataset.

for user $u$, where $L$ is the number of such **historical** trajectories, the goal is to recommend the POIs that $u$ is most likely to visit next after the current **contextual** trajectory $T'_u(t)$.

**POI Recommendation Task Definition**     Given a current contextual trajectory $T'_u(t)$ of user $u$ up to time $t$, along with their historical trajectories $\mathcal{T}_u$, the task of next POI recommendation is to rank all candidate POIs $p_i \in \mathcal{P}$ according to the model's predicted probability that user $u$ will visit each POI next.

Formally, the model learns a ranking function:

$$f : (T'_u(t), \mathcal{T}_u) \to \{\hat{y}_i\}_{i=1}^{|\mathcal{P}|}$$

where $\hat{y}_i$ denotes the predicted likelihood that user $u$ will visit POI $p_i$ next. Based on these scores, a ranked list of POIs is returned as recommendations.

This formulation enables POI recommendation, where the goal is to suggest a set of likely POIs that a user may visit next, based on their historical check-ins and inferred preferences. Our evaluation metrics, Acc@k and NDCG@k, assess whether the ground-truth POI appears among the top-$k$ ranked candidates, reflecting the quality of the recommended set. In particular, Acc@1 captures the stricter task of *immediate* next POI prediction, measuring whether the top-ranked POI matches the user's actual next visit.

## C.2    MODELS

For thoroughness, we evaluated the following models as baselines:

- **FPMC** (Rendle et al., 2010): A classical baseline that combines first-order Markov chains with matrix factorization to model personalized next-location predictions.
- **RNN** (Wang et al., 2021a), **LSTPM** (Sun et al., 2020), and **DeepMove** (Feng et al., 2018): Recurrent neural networks designed to capture sequential dependencies, with varying mechanisms to incorporate spatio-temporal context.
- **GETNext** (Yang et al., 2022c) and **STHGCN** (Yan et al., 2023): Transformer-based graph neural networks to model social, spatial, and temporal dependencies.
- **UniMove** (Han et al., 2025b): Trajectory foundation model based on a Transformer decoder architecture with Mixture of Experts (MoE) layers.

## C.3    EXPERIMENT AND IMPLEMENTATION DETAILS

For training and evaluation, we used the LibCity[1] library (Wang et al., 2021a), which provides implementations of classical baselines including FPMC (Rendle et al., 2010), RNN (Wang et al., 2021a), LSTPM (Sun et al., 2020), and DeepMove (Feng et al., 2018). The training hyperparameters are listed in Table 10 and, unless otherwise noted, follow the default configurations provided by LibCity.

For GETNext[2] (Yang et al., 2022c) and STHGCN[3] (Yan et al., 2023), we adapted the original source code released by the respective authors. Due to variations in dataset sizes and training costs across cities, we applied different hyperparameters for some cities, as detailed in Table 11.

For UniMove[4] (Han et al., 2025b), we modified their original source code for Massive-STEPS. For location features, we used Schema.org's 162 list of categories as a categorical feature and the administrative region as the grid area for POI category distribution. Hyperparameters are listed in Table 10 and, unless otherwise noted, follow the default values.

All modified code implementations are available as submodules in our main dataset repository. Experiments were conducted using NVIDIA L4, L40S, and H100 GPUs.

---

[1]`https://github.com/libcity/bigscity-libcity-datasets/`

[2]`https://github.com/songyangme/GETNext`

[3]`https://github.com/alipay/Spatio-Temporal-Hypergraph-Model`

[4]`https://github.com/tsinghua-fib-lab/unimove`

Table 10: **Hyperparameters for Markov-based methods, recurrent networks, and UniMove**.

| Hyperparameter | FPMC | RNN | LSTPM | DeepMove | UniMove |
|---|---|---|---|---|---|
| Batch Size | 20 | 20 | 20 | 20 | 4 |
| Learning Rate | 5e-4 | 1e-3 | 1e-4 | 1e-3 | 3e-4 |
| Max Epoch | 1 | 30 | 40 | 30 | 50 |
| Location Embedding Size | 64 | 500 | 500 | 500 | {256, 128} |
| Hidden Embedding Size | N/A | 500 | 500 | 500 | 512 |
| Dropout | N/A | 0.3 | 0.8 | 0.5 | N/A |

Table 11: **Hyperparameters for Transformer-based graph neural networks**.

| Model | Cities | Batch Size | LR | Epochs |
|---|---|---|---|---|
| GETNext | Beijing, Melbourne, Moscow, Palembang, Shanghai, Sydney, Tokyo | 16 | 1e-3 | 200 |
| | Bandung, Istanbul, Kuwait City, New York, Petaling Jaya, São Paulo, Tangerang | 16 | 1e-4 | 20 |
| | Jakarta | 16 | 5e-5 | 20 |
| STHGCN | Beijing, Melbourne, Palembang, Shanghai, Sydney, Tokyo | 16 | 1e-4 | 20 |
| | Bandung, Istanbul, Jakarta, Kuwait City, Moscow, New York, Petaling Jaya, São Paulo, Tangerang | 64 | 1e-4 | 20 |

## C.4 SUPPLEMENTARY RESULTS

We report the full results of our supervised POI recommendation baselines in Table 12, 13 and 14, using three evaluation metrics: Acc@1, Acc@5, and NDCG@5.

Table 12: **Performance of supervised POI recommendation baselines across 5 cities**: Bandung, Beijing, Istanbul, Jakarta, Kuwait City. We report three metrics: Acc@1 (A@1), Acc@5 (A@5), and NDCG@5 (N@5).

| Model | Bandung | | | Beijing | | | Istanbul | | | Jakarta | | | Kuwait City | | |
|---|---|---|---|---|---|---|---|---|---|---|---|---|---|---|---|
| | A@1 | A@5 | N@5 | A@1 | A@5 | N@5 | A@1 | A@5 | N@5 | A@1 | A@5 | N@5 | A@1 | A@5 | N@5 |
| FPMC | 0.048 | 0.118 | 0.083 | 0.000 | 0.021 | 0.009 | 0.026 | 0.074 | 0.050 | 0.029 | 0.085 | 0.058 | 0.021 | 0.089 | 0.054 |
| RNN | 0.062 | 0.135 | 0.099 | 0.085 | 0.183 | 0.134 | 0.077 | 0.178 | 0.130 | 0.049 | 0.115 | 0.083 | 0.087 | 0.203 | 0.146 |
| LSTPM | 0.110 | 0.241 | 0.179 | 0.127 | 0.211 | 0.169 | 0.142 | 0.286 | 0.217 | 0.099 | 0.210 | 0.157 | 0.180 | 0.362 | 0.275 |
| DeepMove | 0.107 | 0.232 | 0.172 | 0.106 | 0.261 | 0.190 | 0.150 | 0.298 | 0.228 | 0.103 | 0.212 | 0.160 | 0.179 | 0.360 | 0.274 |
| GETNext | 0.179 | 0.306 | 0.247 | 0.433 | 0.527 | 0.486 | 0.146 | 0.268 | 0.210 | 0.155 | 0.257 | 0.209 | 0.175 | 0.322 | 0.251 |
| STHGCN | 0.219 | 0.375 | 0.302 | 0.453 | 0.640 | 0.552 | 0.241 | 0.385 | 0.318 | 0.197 | 0.334 | 0.270 | 0.225 | 0.394 | 0.314 |
| UniMove | 0.007 | 0.060 | 0.033 | 0.036 | 0.205 | 0.128 | 0.015 | 0.061 | 0.038 | 0.004 | 0.036 | 0.020 | 0.023 | 0.120 | 0.073 |

# D ANALYZING URBAN FEATURES AND POI RECOMMENDATION PERFORMANCE

As discussed in Section 2.2, several hypotheses have been proposed to explain why POI recommendation models perform better in certain cities than others. These hypotheses aim to uncover how various urban features affect model performance. For example, Gowalla-CA (Cho et al., 2011; Yuan et al., 2013) often yields lower accuracy compared to FSQ-NYC and FSQ-TKY (Yang et al., 2014), suggesting that some cities may be inherently harder to model. In this analysis, we focus on supervised models only.

Prior studies (Yang et al., 2022c; Yan et al., 2023; Li et al., 2024) have suggested several features as potential explanatory variables, including:

- Number of unique check-ins,
- Number of unique trajectories,
- Number of unique POI categories,
- Geographical area (larger areas are assumed to be harder to model), and
- POI density or spatial sparsity (i.e., unique POIs per unit area).

Table 13: **Performance of supervised POI recommendation baselines across 5 cities**: Melbourne, Moscow, New York, Palembang, Petaling Jaya. We report three metrics: Acc@1 (A@1), Acc@5 (A@5), and NDCG@5 (N@5).

| Model | Melbourne | | | Moscow | | | New York | | | Palembang | | | Petaling Jaya | | |
|---|---|---|---|---|---|---|---|---|---|---|---|---|---|---|---|
| | A@1 | A@5 | N@5 | A@1 | A@5 | N@5 | A@1 | A@5 | N@5 | A@1 | A@5 | N@5 | A@1 | A@5 | N@5 |
| FPMC | 0.062 | 0.147 | 0.107 | 0.059 | 0.129 | 0.094 | 0.032 | 0.090 | 0.061 | 0.102 | 0.169 | 0.136 | 0.026 | 0.084 | 0.057 |
| RNN | 0.059 | 0.105 | 0.083 | 0.075 | 0.164 | 0.122 | 0.061 | 0.119 | 0.092 | 0.049 | 0.121 | 0.085 | 0.064 | 0.148 | 0.107 |
| LSTPM | 0.091 | 0.204 | 0.150 | 0.151 | 0.300 | 0.229 | 0.099 | 0.206 | 0.155 | 0.114 | 0.230 | 0.175 | 0.099 | 0.222 | 0.163 |
| DeepMove | 0.083 | 0.179 | 0.134 | 0.143 | 0.283 | 0.217 | 0.097 | 0.195 | 0.149 | 0.084 | 0.191 | 0.139 | 0.112 | 0.234 | 0.175 |
| GETNext | 0.100 | 0.250 | 0.179 | 0.175 | 0.335 | 0.260 | 0.134 | 0.263 | 0.202 | 0.158 | 0.313 | 0.239 | 0.139 | 0.254 | 0.200 |
| STHGCN | 0.168 | 0.318 | 0.247 | 0.223 | 0.382 | 0.308 | 0.146 | 0.259 | 0.207 | 0.246 | 0.427 | 0.341 | 0.174 | 0.301 | 0.241 |
| UniMove | 0.008 | 0.066 | 0.037 | 0.009 | 0.051 | 0.030 | 0.004 | 0.028 | 0.016 | 0.009 | 0.060 | 0.035 | 0.008 | 0.058 | 0.034 |

Table 14: **Performance of supervised POI recommendation baselines across 5 cities**: São Paulo, Shanghai, Sydney, Tangerang, Tokyo. We report three metrics: Acc@1 (A@1), Acc@5 (A@5), and NDCG@5 (N@5).

| Model | São Paulo | | | Shanghai | | | Sydney | | | Tangerang | | | Tokyo | | |
|---|---|---|---|---|---|---|---|---|---|---|---|---|---|---|---|
| | A@1 | A@5 | N@5 | A@1 | A@5 | N@5 | A@1 | A@5 | N@5 | A@1 | A@5 | N@5 | A@1 | A@5 | N@5 |
| FPMC | 0.030 | 0.079 | 0.055 | 0.084 | 0.154 | 0.120 | 0.075 | 0.180 | 0.131 | 0.104 | 0.220 | 0.166 | 0.176 | 0.291 | 0.239 |
| RNN | 0.097 | 0.191 | 0.147 | 0.055 | 0.120 | 0.090 | 0.080 | 0.164 | 0.125 | 0.087 | 0.179 | 0.135 | 0.133 | 0.254 | 0.197 |
| LSTPM | 0.158 | 0.319 | 0.243 | 0.099 | 0.195 | 0.149 | 0.141 | 0.265 | 0.206 | 0.154 | 0.309 | 0.237 | 0.225 | 0.394 | 0.315 |
| DeepMove | 0.160 | 0.310 | 0.240 | 0.085 | 0.168 | 0.128 | 0.129 | 0.240 | 0.188 | 0.145 | 0.285 | 0.219 | 0.201 | 0.362 | 0.288 |
| GETNext | 0.202 | 0.360 | 0.286 | 0.115 | 0.230 | 0.177 | 0.181 | 0.347 | 0.266 | 0.224 | 0.372 | 0.302 | 0.180 | 0.361 | 0.275 |
| STHGCN | 0.250 | 0.425 | 0.344 | 0.193 | 0.329 | 0.264 | 0.227 | 0.378 | 0.307 | 0.293 | 0.492 | 0.400 | 0.250 | 0.432 | 0.350 |
| UniMove | 0.002 | 0.018 | 0.009 | 0.000 | 0.055 | 0.029 | 0.015 | 0.102 | 0.059 | 0.001 | 0.055 | 0.029 | 0.032 | 0.109 | 0.072 |

We also propose several additional features for consideration:

- Number of unique POIs,
- Check-in density (unique check-ins per area),
- Trajectory density (unique trajectories per area), and
- Category entropy, our proposed feature capturing category diversity.

**Category entropy**, based on Shannon entropy, measures how evenly POI categories are distributed in a city. A higher entropy suggests that check-ins are spread across a wide variety of categories, while a lower entropy indicates a concentration in fewer types. The formula for Shannon entropy is:

$$H = -\sum_{i=1}^{N} p_i \log(p_i) \tag{1}$$

where $p_i$ is the proportion of venues in category $i$, and $N$ is the total number of POI categories. The proportion $p_i$ is defined as:

$$p_i = \frac{c_i}{\sum_{j=1}^{N} c_j} \tag{2}$$

where $c_i$ is the count of venues in category $i$. In other words, $p_i$ represents the fraction of all venues that belong to category $i$.

Moreover, previous studies have primarily focused on only three datasets: FSQ-NYC, FSQ-TKY, and Gowalla-CA. In contrast, Massive-STEPS provides broader coverage across 15 cities, enabling a more comprehensive and robust analysis. To examine the relationship between urban features and model performance, we averaged the three evaluation metrics across six supervised baselines for each city and computed the Spearman correlation with each candidate feature. To further support our findings, we also included the results of GETNext (Yang et al., 2022c) and STHGCN (Yan et al., 2023) on FSQ-NYC, FSQ-TKY, and Gowalla-CA, calculated their corresponding urban features, and

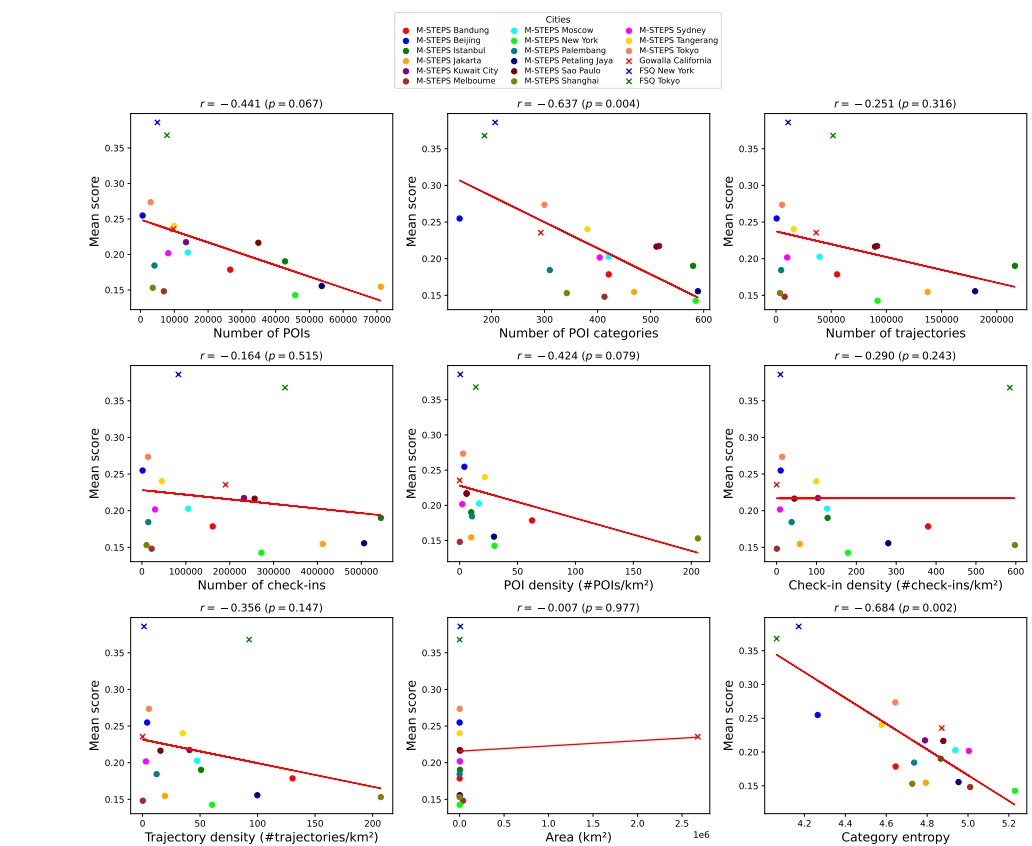

Figure 9: **Spearman correlation between nine candidate urban features and the mean score of POI recommendation models** across 15 cities, including Massive-STEPS (ours), FSQ (Yang et al., 2014), and Gowalla (Cho et al., 2011; Yuan et al., 2013).

averaged the reported metrics of each city. Fig. 9 presents the correlations between all nine features and the average performance metric.

Among all features, **category entropy** shows the strongest correlation with model performance, with a Spearman correlation of $r = -0.684$ ($p = 0.002$). This suggests that cities with more evenly distributed POI categories *tend* to yield lower prediction accuracy. Intuitively, when no single category dominates (a city has roughly equal proportions of restaurants, cafes, homes, and other POIs), it becomes more difficult for models to predict a user's next destination. In these cases, user behavior is more varied and less predictable. On the other hand, cities with more skewed category distributions (e.g., mostly food places or mostly residential areas) tend to have more consistent patterns of movement, making them easier for models to learn and predict. Interestingly, our finding contradicts the hypothesis proposed by LLM4POI (Li et al., 2024), which suggests that FSQ-NYC is easier to model than Gowalla-CA due to the former's vast number of POI categories, which were supposed to provide richer contextual signals for the model. Our results indicate that it is not the number of categories that matters, but rather how these categories are distributed.

# E   ZERO-SHOT POI RECOMMENDATION: TASK DETAILS

## E.1   PROBLEM FORMULATION

The zero-shot POI recommendation task follows the same problem formulation as its supervised counterpart (see Section C.1). The key difference is that in this setting, the model parameters remain frozen and the models are pre-trained, rather than trained from randomly initialized weights.

### E.2 METHODS

We evaluated three LLM-based prompting methods:

- **LLM-Mob** (Wang et al., 2023c): One of the earliest methods to use LLMs for next POI prediction, prompting LLMs with both historical and current (contextual) trajectories.
- **LLM-ZS** (Beneduce et al., 2024): A simplified version of LLM-Mob that retains the use of historical and contextual trajectories but simplifies its prompt design.
- **LLM-Move** (Feng et al., 2024a): Extends previous prompting methods by introducing a RAG-like approach, retrieving nearby POIs as candidates, and ranking them by geographic distance to the user's most recent visit.

### E.3 EXPERIMENT AND IMPLEMENTATION DETAILS

**Preprocessing**  We adopted the AgentMove[5] library (Feng et al., 2025), which provides implementations of three LLM methods: LLM-Mob (Wang et al., 2023c), LLM-ZS (Beneduce et al., 2024), and LLM-Move (Feng et al., 2024a). The preprocessing steps used by AgentMove are as follows.

First, we selected 200 random users from the test set and sampled one random trajectory for each user. This trajectory serves as the **context stays**, representing the current trajectory to be predicted. The **historical stays** are composed of the most recent 15 trajectories from the same user, drawn from the training set. Each check-in is described by four attributes: the hour (in 12-hour format), the day of the week, the POI ID, and the POI category name.

Second, the LLMs are set to return outputs in JSON format, generating the top 5 predicted POI IDs along with an explanation of their reasoning. Following the AgentMove setup and to ensure replicability, we set the generation parameters as follows: a temperature of 0.0, a maximum output length of 1000 tokens, and an input context window capped at 2000 tokens.

**Prompting**  Prompt templates for each method, LLM-Mob, LLM-ZS, and LLM-Move, are presented in Listing 1, 2, and 3, respectively.

```
1 Your task is to predict a user's next location based on his/her activity
     pattern.
2 You will be provided with <history> which is a list containing this user'
     s historical stays, then <context> which provide contextual
     information
3 about where and when this user has been to recently. Stays in both <
     history> and <context> are in chronological order.
4 Each stay takes on such form as (start_time, day_of_week, duration,
     place_id). The detailed explanation of each element is as follows:
5 start_time: the start time of the stay in 12h clock format.
6 day_of_week: indicating the day of the week.
7 duration: an integer indicating the duration (in minute) of each stay.
     Note that this will be None in the <target_stay> introduced later.
8 place_id: an integer representing the unique place ID, which indicates
     where the stay is.
9
10 Then you need to do next location prediction on <target_stay> which is
     the prediction target with unknown place ID denoted as <next_place_id
     > and
11 unknown duration denoted as None, while temporal information is provided.
12
13 Please infer what the <next_place_id> might be (please output the 10 most
      likely places which are ranked in descending order in terms of
     probability), considering the following aspects:
14 1. the activity pattern of this user that you learned from <history>, e.g
     ., repeated visits to certain places during certain times;
15 2. the context stays in <context>, which provide more recent activities
     of this user;
```

[5]https://github.com/tsinghua-fib-lab/agentmove/

```
16 3. the temporal information (i.e., start_time and day_of_week) of target
       stay, which is important because people's activity varies during
       different time (e.g., nighttime versus daytime)
17 and on different days (e.g., weekday versus weekend).
18
19 Please organize your answer in a JSON object containing following keys:
20 "prediction" (the ID of the five most probable places in descending order
       of probability) and "reason" (a concise explanation that supports
       your prediction). Do not include line breaks in your output.
21
22 The data are as follows:
23 <history>: {historical_stays}
24 <context>: {context_stays}
25 <target_stay>: {target_time, target_day_of_week}
```

Listing 1: Prompt for LLM-Mob

```
1 Your task is to predict <next_place_id> in <target_stay>, a location with
       an unknown ID, while temporal data is available.
2
3 Predict <next_place_id> by considering:
4 1. The user's activity trends gleaned from <historical_stays> and the
       current activities from  <context_stays>.
5 2. Temporal details (start_time and day_of_week) of the target stay,
       crucial for understanding activity variations.
6
7 Present your answer in a JSON object with:
8 "prediction" (IDs of the five most probable places, ranked by probability
       ) and "reason" (a concise justification for your prediction).
9
10 The data:
11 <historical_stays>: {historical_stays}
12 <context_stays>: {context_stays}
13 <target_stay>: {target_time, target_day_of_week}
```

Listing 2: Prompt for LLM-ZS

```
1 <long-term check-ins> [Format: (POIID, Category)]: {historical_stays}
2 <recent check-ins> [Format: (POIID, Category)]: {context_stays}
3 <candidate set> [Format: (POIID, Distance, Category)]: {candidates}
4 Your task is to recommend a user's next point-of-interest (POI) from <
       candidate set> based on his/her trajectory information.
5 The trajectory information is made of a sequence of the user's <long-term
       check-ins> and a sequence of the user's <recent check-ins> in
       chronological order.
6 Now I explain the elements in the format. "POIID" refers to the unique id
       of the POI, "Distance" indicates the distance (kilometers) between
       the user and the POI, and "Category" shows the semantic information
       of the POI.
7
8 Requirements:
9 1. Consider the long-term check-ins to extract users' long-term
       preferences since people tend to revisit their frequent visits.
10 2. Consider the recent check-ins to extract users' current perferences.
11 3. Consider the "Distance" since people tend to visit nearby pois.
12 4. Consider which "Category" the user would go next for long-term check-
       ins indicates sequential transitions the user prefer.
13
14 Please organize your answer in a JSON object containing following keys:
15 "prediction" (10 distinct POIIDs of the ten most probable places in <
       candidate set> in descending order of probability), and "reason" (a
       concise explanation that supports your recommendation according to
       the requirements). Do not include line breaks in your output.
```

Listing 3: Prompt for LLM-Move

**Models and Implementations**    We use the following LLMs in our experiments:

- Gemini 2.0 Flash (`gemini-2.0-flash`),

- Qwen 2.5 7B Instruct (`Qwen2.5-7B-Instruct-AWQ`)[6],

- Llama 3.1 8B Instruct (`Meta-Llama-3.1-8B-Instruct-AWQ-INT4`)[7],

- Gemma 2 9B Instruct (`gemma-2-9b-it-AWQ-INT4`)[8].

All open-source models are quantized using AWQ (Lin et al., 2024) and served via vLLM (Kwon et al., 2023). Inference of open-source models was conducted on NVIDIA A100 GPUs. We accessed Gemini via the official API. All modified code implementations are publicly available in our main dataset repository.

### E.4    SUPPLEMENTARY RESULTS

We provide the full results of our zero-shot POI recommendation results in Table 15, 16, and 17, providing three metrics: Acc@1, Acc@5, and NDCG@5. Additionally, Table 18, 19, and 20 present zero-shot performance across two time periods (2012-2013 and 2017-2018) using LLM-Move.

Table 15: **Performance of zero-shot POI recommendation baselines across 5 cities**: Bandung, Beijing, Istanbul, Jakarta, Kuwait City. We report three metrics: Acc@1 (A@1), Acc@5 (A@5), and NDCG@5 (N@5).

| Method | Model | Bandung | | | Beijing | | | Istanbul | | | Jakarta | | | Kuwait City | | |
|---|---|---|---|---|---|---|---|---|---|---|---|---|---|---|---|---|
| | | A@1 | A@5 | N@5 | A@1 | A@5 | N@5 | A@1 | A@5 | N@5 | A@1 | A@5 | N@5 | A@1 | A@5 | N@5 |
| LLM-Mob | Gemini 2 Flash | 0.105 | 0.170 | 0.139 | 0.115 | 0.308 | 0.226 | 0.080 | 0.225 | 0.160 | 0.100 | 0.245 | 0.174 | 0.095 | 0.270 | 0.185 |
| | Qwen 2.5 7B | 0.060 | 0.155 | 0.111 | 0.058 | 0.385 | 0.218 | 0.035 | 0.240 | 0.148 | 0.105 | 0.245 | 0.179 | 0.080 | 0.220 | 0.155 |
| | Llama 3.1 8B | 0.010 | 0.100 | 0.055 | 0.000 | 0.000 | 0.000 | 0.020 | 0.110 | 0.065 | 0.055 | 0.150 | 0.104 | 0.030 | 0.100 | 0.066 |
| | Gemma 2 9B | 0.070 | 0.175 | 0.126 | 0.115 | 0.288 | 0.206 | 0.075 | 0.200 | 0.146 | 0.105 | 0.240 | 0.178 | 0.080 | 0.210 | 0.150 |
| LLM-ZS | Gemini 2 Flash | 0.095 | 0.195 | 0.147 | 0.058 | 0.385 | 0.246 | 0.090 | 0.235 | 0.166 | 0.110 | 0.250 | 0.188 | 0.080 | 0.245 | 0.167 |
| | Qwen 2.5 7B | 0.055 | 0.185 | 0.126 | 0.038 | 0.404 | 0.237 | 0.040 | 0.235 | 0.141 | 0.065 | 0.250 | 0.161 | 0.050 | 0.220 | 0.140 |
| | Llama 3.1 8B | 0.045 | 0.210 | 0.131 | 0.077 | 0.346 | 0.221 | 0.040 | 0.225 | 0.137 | 0.045 | 0.200 | 0.126 | 0.060 | 0.210 | 0.137 |
| | Gemma 2 9B | 0.065 | 0.185 | 0.130 | 0.096 | 0.308 | 0.217 | 0.045 | 0.225 | 0.141 | 0.105 | 0.250 | 0.180 | 0.070 | 0.230 | 0.153 |
| LLM-Move | Gemini 2 Flash | 0.225 | 0.350 | 0.289 | 0.096 | 0.346 | 0.218 | 0.205 | 0.385 | 0.289 | 0.295 | 0.405 | 0.350 | 0.220 | 0.380 | 0.295 |
| | Qwen 2.5 7B | 0.100 | 0.155 | 0.128 | 0.192 | 0.346 | 0.280 | 0.175 | 0.270 | 0.226 | 0.115 | 0.225 | 0.169 | 0.160 | 0.285 | 0.227 |
| | Llama 3.1 8B | 0.030 | 0.035 | 0.033 | 0.058 | 0.135 | 0.100 | 0.015 | 0.055 | 0.036 | 0.015 | 0.025 | 0.021 | 0.010 | 0.035 | 0.023 |
| | Gemma 2 9B | 0.175 | 0.245 | 0.213 | 0.096 | 0.365 | 0.229 | 0.100 | 0.200 | 0.155 | 0.235 | 0.290 | 0.266 | 0.120 | 0.275 | 0.202 |

Table 16: **Performance of zero-shot POI recommendation baselines across 5 cities**: Melbourne, Moscow, New York, Palembang, Petaling Jaya. We report three metrics: Acc@1 (A@1), Acc@5 (A@5), and NDCG@5 (N@5).

| Method | Model | Melbourne | | | Moscow | | | New York | | | Palembang | | | Petaling Jaya | | |
|---|---|---|---|---|---|---|---|---|---|---|---|---|---|---|---|---|
| | | A@1 | A@5 | N@5 | A@1 | A@5 | N@5 | A@1 | A@5 | N@5 | A@1 | A@5 | N@5 | A@1 | A@5 | N@5 |
| LLM-Mob | Gemini 2 Flash | 0.060 | 0.150 | 0.108 | 0.130 | 0.245 | 0.187 | 0.095 | 0.175 | 0.136 | 0.135 | 0.275 | 0.208 | 0.090 | 0.220 | 0.160 |
| | Qwen 2.5 7B | 0.030 | 0.130 | 0.083 | 0.090 | 0.270 | 0.185 | 0.070 | 0.185 | 0.131 | 0.075 | 0.205 | 0.143 | 0.030 | 0.195 | 0.116 |
| | Llama 3.1 8B | 0.010 | 0.065 | 0.040 | 0.030 | 0.100 | 0.068 | 0.025 | 0.090 | 0.061 | 0.005 | 0.040 | 0.025 | 0.010 | 0.090 | 0.050 |
| | Gemma 2 9B | 0.055 | 0.150 | 0.108 | 0.100 | 0.240 | 0.176 | 0.070 | 0.175 | 0.124 | 0.095 | 0.240 | 0.171 | 0.055 | 0.185 | 0.122 |
| LLM-ZS | Gemini 2 Flash | 0.065 | 0.160 | 0.115 | 0.125 | 0.300 | 0.217 | 0.080 | 0.170 | 0.129 | 0.130 | 0.260 | 0.196 | 0.110 | 0.210 | 0.164 |
| | Qwen 2.5 7B | 0.040 | 0.155 | 0.100 | 0.080 | 0.260 | 0.176 | 0.050 | 0.180 | 0.116 | 0.050 | 0.215 | 0.135 | 0.045 | 0.175 | 0.111 |
| | Llama 3.1 8B | 0.040 | 0.155 | 0.101 | 0.080 | 0.270 | 0.183 | 0.055 | 0.160 | 0.111 | 0.070 | 0.240 | 0.154 | 0.030 | 0.205 | 0.123 |
| | Gemma 2 9B | 0.050 | 0.140 | 0.100 | 0.080 | 0.290 | 0.194 | 0.075 | 0.185 | 0.129 | 0.060 | 0.235 | 0.150 | 0.065 | 0.185 | 0.126 |
| LLM-Move | Gemini 2 Flash | 0.225 | 0.325 | 0.275 | 0.220 | 0.400 | 0.316 | 0.235 | 0.415 | 0.325 | 0.260 | 0.385 | 0.329 | 0.210 | 0.335 | 0.273 |
| | Qwen 2.5 7B | 0.110 | 0.220 | 0.165 | 0.230 | 0.310 | 0.274 | 0.120 | 0.255 | 0.188 | 0.130 | 0.195 | 0.163 | 0.135 | 0.175 | 0.155 |
| | Llama 3.1 8B | 0.040 | 0.195 | 0.123 | 0.005 | 0.065 | 0.031 | 0.035 | 0.130 | 0.084 | 0.010 | 0.015 | 0.013 | 0.040 | 0.060 | 0.049 |
| | Gemma 2 9B | 0.115 | 0.275 | 0.199 | 0.110 | 0.245 | 0.185 | 0.115 | 0.245 | 0.183 | 0.210 | 0.270 | 0.240 | 0.175 | 0.235 | 0.208 |

---

[6]`https://huggingface.co/qwen/qwen2.5-7b-instruct-awq`
[7]`https://huggingface.co/hugging-quants/Meta-Llama-3.1-8B-Instruct-AWQ-INT4`
[8]`https://huggingface.co/hugging-quants/gemma-2-9b-it-AWQ-INT4`

Table 17: **Performance of zero-shot POI recommendation baselines across 5 cities**: São Paulo, Shanghai, Sydney, Tangerang, Tokyo. We report three metrics: Acc@1 (A@1), Acc@5 (A@5), and NDCG@5 (N@5).

| Method | Model | São Paulo | | | Shanghai | | | Sydney | | | Tangerang | | | Tokyo | | |
|---|---|---|---|---|---|---|---|---|---|---|---|---|---|---|---|---|
| | | A@1 | A@5 | N@5 | A@1 | A@5 | N@5 | A@1 | A@5 | N@5 | A@1 | A@5 | N@5 | A@1 | A@5 | N@5 |
| LLM-Mob | Gemini 2 Flash | 0.130 | 0.305 | 0.223 | 0.055 | 0.160 | 0.111 | 0.060 | 0.160 | 0.112 | 0.155 | 0.285 | 0.225 | 0.140 | 0.320 | 0.238 |
| | Qwen 2.5 7B | 0.090 | 0.290 | 0.188 | 0.040 | 0.170 | 0.108 | 0.035 | 0.145 | 0.091 | 0.095 | 0.285 | 0.196 | 0.110 | 0.350 | 0.243 |
| | Llama 3.1 8B | 0.030 | 0.165 | 0.098 | 0.005 | 0.020 | 0.013 | 0.020 | 0.085 | 0.053 | 0.020 | 0.120 | 0.073 | 0.005 | 0.045 | 0.025 |
| | Gemma 2 9B | 0.085 | 0.230 | 0.162 | 0.050 | 0.150 | 0.104 | 0.030 | 0.130 | 0.086 | 0.145 | 0.270 | 0.209 | 0.145 | 0.345 | 0.255 |
| LLM-ZS | Gemini 2 Flash | 0.150 | 0.315 | 0.235 | 0.065 | 0.160 | 0.113 | 0.060 | 0.155 | 0.111 | 0.145 | 0.310 | 0.234 | 0.160 | 0.380 | 0.278 |
| | Qwen 2.5 7B | 0.095 | 0.290 | 0.198 | 0.045 | 0.155 | 0.103 | 0.045 | 0.170 | 0.109 | 0.100 | 0.315 | 0.215 | 0.120 | 0.365 | 0.257 |
| | Llama 3.1 8B | 0.030 | 0.280 | 0.159 | 0.060 | 0.165 | 0.116 | 0.040 | 0.185 | 0.110 | 0.080 | 0.255 | 0.173 | 0.110 | 0.415 | 0.269 |
| | Gemma 2 9B | 0.075 | 0.300 | 0.192 | 0.050 | 0.165 | 0.112 | 0.045 | 0.155 | 0.103 | 0.100 | 0.330 | 0.227 | 0.110 | 0.395 | 0.263 |
| LLM-Move | Gemini 2 Flash | 0.285 | 0.415 | 0.350 | 0.170 | 0.270 | 0.221 | 0.230 | 0.420 | 0.331 | 0.200 | 0.340 | 0.274 | 0.250 | 0.470 | 0.368 |
| | Qwen 2.5 7B | 0.155 | 0.235 | 0.199 | 0.095 | 0.165 | 0.133 | 0.125 | 0.280 | 0.205 | 0.175 | 0.280 | 0.229 | 0.250 | 0.360 | 0.312 |
| | Llama 3.1 8B | 0.045 | 0.045 | 0.045 | 0.020 | 0.040 | 0.030 | 0.055 | 0.220 | 0.141 | 0.000 | 0.005 | 0.003 | 0.030 | 0.060 | 0.046 |
| | Gemma 2 9B | 0.195 | 0.300 | 0.252 | 0.105 | 0.150 | 0.128 | 0.125 | 0.370 | 0.254 | 0.125 | 0.250 | 0.193 | 0.130 | 0.305 | 0.225 |

Table 18: **Performance of zero-shot POI recommendation using LLM-Move across two time periods and 5 cities**: Bandung, Beijing, Istanbul, Jakarta, and Kuwait City. We report three metrics: Acc@1 (A@1), Acc@5 (A@5), and NDCG@5 (N@5).

| Time Period | Model | Bandung | | | Beijing | | | Istanbul | | | Jakarta | | | Kuwait City | | |
|---|---|---|---|---|---|---|---|---|---|---|---|---|---|---|---|---|
| | | A@1 | A@5 | N@5 | A@1 | A@5 | N@5 | A@1 | A@5 | N@5 | A@1 | A@5 | N@5 | A@1 | A@5 | N@5 |
| 2012-2013 | Gemini 2 Flash | 0.227 | 0.351 | 0.290 | 0.102 | 0.367 | 0.232 | 0.212 | 0.384 | 0.290 | 0.295 | 0.409 | 0.352 | 0.423 | 0.500 | 0.453 |
| | Qwen 2.5 7B | 0.098 | 0.155 | 0.126 | 0.204 | 0.367 | 0.298 | 0.192 | 0.295 | 0.247 | 0.114 | 0.223 | 0.167 | 0.269 | 0.423 | 0.357 |
| | Llama 3.1 8B | 0.031 | 0.036 | 0.034 | 0.041 | 0.122 | 0.086 | 0.007 | 0.027 | 0.017 | 0.010 | 0.021 | 0.016 | 0.000 | 0.000 | 0.000 |
| | Gemma 2 9B | 0.180 | 0.247 | 0.217 | 0.102 | 0.388 | 0.244 | 0.116 | 0.199 | 0.159 | 0.228 | 0.285 | 0.260 | 0.308 | 0.385 | 0.342 |
| 2017-2018 | Gemini 2 Flash | 0.167 | 0.333 | 0.272 | 0.000 | 0.000 | 0.000 | 0.185 | 0.389 | 0.284 | 0.286 | 0.286 | 0.286 | 0.190 | 0.362 | 0.271 |
| | Qwen 2.5 7B | 0.167 | 0.167 | 0.167 | 0.000 | 0.000 | 0.000 | 0.130 | 0.204 | 0.168 | 0.143 | 0.286 | 0.233 | 0.144 | 0.264 | 0.208 |
| | Llama 3.1 8B | 0.000 | 0.000 | 0.000 | 0.333 | 0.333 | 0.333 | 0.037 | 0.130 | 0.088 | 0.143 | 0.143 | 0.143 | 0.011 | 0.040 | 0.026 |
| | Gemma 2 9B | 0.000 | 0.167 | 0.083 | 0.000 | 0.000 | 0.000 | 0.056 | 0.204 | 0.142 | 0.429 | 0.429 | 0.429 | 0.092 | 0.259 | 0.181 |

Table 19: **Performance of zero-shot POI recommendation using LLM-Move across two time periods and 5 cities**: Melbourne, Moscow, New York, Palembang, Petaling Jaya. We report three metrics: Acc@1 (A@1), Acc@5 (A@5), and NDCG@5 (N@5).

| Time Period | Model | Melbourne | | | Moscow | | | New York | | | Palembang | | | Petaling Jaya | | |
|---|---|---|---|---|---|---|---|---|---|---|---|---|---|---|---|---|
| | | A@1 | A@5 | N@5 | A@1 | A@5 | N@5 | A@1 | A@5 | N@5 | A@1 | A@5 | N@5 | A@1 | A@5 | N@5 |
| 2012-2013 | Gemini 2 Flash | 0.226 | 0.329 | 0.279 | 0.218 | 0.401 | 0.316 | 0.240 | 0.403 | 0.320 | 0.256 | 0.385 | 0.327 | 0.199 | 0.348 | 0.274 |
| | Qwen 2.5 7B | 0.116 | 0.232 | 0.175 | 0.234 | 0.310 | 0.275 | 0.130 | 0.240 | 0.190 | 0.128 | 0.195 | 0.162 | 0.142 | 0.184 | 0.164 |
| | Llama 3.1 8B | 0.039 | 0.200 | 0.125 | 0.005 | 0.066 | 0.032 | 0.032 | 0.117 | 0.074 | 0.010 | 0.015 | 0.013 | 0.014 | 0.043 | 0.028 |
| | Gemma 2 9B | 0.097 | 0.271 | 0.189 | 0.112 | 0.249 | 0.188 | 0.130 | 0.260 | 0.198 | 0.215 | 0.272 | 0.244 | 0.199 | 0.262 | 0.234 |
| 2017-2018 | Gemini 2 Flash | 0.222 | 0.311 | 0.261 | 0.333 | 0.333 | 0.333 | 0.217 | 0.457 | 0.342 | 0.400 | 0.400 | 0.400 | 0.237 | 0.305 | 0.273 |
| | Qwen 2.5 7B | 0.089 | 0.178 | 0.132 | 0.000 | 0.333 | 0.167 | 0.087 | 0.304 | 0.183 | 0.200 | 0.200 | 0.200 | 0.119 | 0.153 | 0.134 |
| | Llama 3.1 8B | 0.044 | 0.178 | 0.116 | 0.000 | 0.000 | 0.000 | 0.043 | 0.174 | 0.116 | 0.000 | 0.000 | 0.000 | 0.102 | 0.102 | 0.102 |
| | Gemma 2 9B | 0.178 | 0.289 | 0.232 | 0.000 | 0.000 | 0.000 | 0.065 | 0.196 | 0.133 | 0.000 | 0.200 | 0.086 | 0.119 | 0.169 | 0.144 |

Table 20: **Performance of zero-shot POI recommendation using LLM-Move across two time periods and 5 cities**: São Paulo, Shanghai, Sydney, Tangerang, Tokyo. We report three metrics: Acc@1 (A@1), Acc@5 (A@5), and NDCG@5 (N@5).

| Time Period | Model | São Paulo | | | Shanghai | | | Sydney | | | Tangerang | | | Tokyo | | |
|---|---|---|---|---|---|---|---|---|---|---|---|---|---|---|---|---|
| | | A@1 | A@5 | N@5 | A@1 | A@5 | N@5 | A@1 | A@5 | N@5 | A@1 | A@5 | N@5 | A@1 | A@5 | N@5 |
| 2012-2013 | Gemini 2 Flash | 0.298 | 0.440 | 0.369 | 0.192 | 0.288 | 0.242 | 0.256 | 0.462 | 0.367 | 0.197 | 0.338 | 0.272 | N/A | N/A | N/A |
| | Qwen 2.5 7B | 0.173 | 0.250 | 0.215 | 0.109 | 0.186 | 0.151 | 0.122 | 0.288 | 0.209 | 0.172 | 0.278 | 0.226 | N/A | N/A | N/A |
| | Llama 3.1 8B | 0.048 | 0.048 | 0.048 | 0.006 | 0.026 | 0.016 | 0.064 | 0.224 | 0.148 | 0.000 | 0.005 | 0.003 | N/A | N/A | N/A |
| | Gemma 2 9B | 0.202 | 0.315 | 0.264 | 0.109 | 0.160 | 0.136 | 0.122 | 0.378 | 0.257 | 0.126 | 0.253 | 0.195 | N/A | N/A | N/A |
| 2017-2018 | Gemini 2 Flash | 0.219 | 0.281 | 0.251 | 0.091 | 0.205 | 0.147 | 0.136 | 0.273 | 0.204 | 0.500 | 0.500 | 0.500 | 0.250 | 0.470 | 0.368 |
| | Qwen 2.5 7B | 0.063 | 0.156 | 0.115 | 0.045 | 0.091 | 0.070 | 0.136 | 0.250 | 0.192 | 0.500 | 0.500 | 0.500 | 0.250 | 0.360 | 0.312 |
| | Llama 3.1 8B | 0.031 | 0.031 | 0.031 | 0.068 | 0.091 | 0.077 | 0.023 | 0.205 | 0.117 | 0.000 | 0.000 | 0.000 | 0.030 | 0.060 | 0.046 |
| | Gemma 2 9B | 0.156 | 0.219 | 0.189 | 0.091 | 0.114 | 0.101 | 0.136 | 0.341 | 0.246 | 0.000 | 0.000 | 0.000 | 0.130 | 0.305 | 0.225 |

# F  Spatiotemporal Classification and Reasoning: Task Details

## F.1  Problem Formulation

Borrowing the notation used in Section C, we formulate this task as follows. Given a current contextual trajectory $T'_u(t)$ of user $u$ up to time $t$, the goal of spatiotemporal trajectory classification is to predict a property $y$ of the trajectory. In this study, we focus on **weekday/weekend classification**, where $y \in \{\text{weekday}, \text{weekend}\}$.

Formally, the LLM serves as a classification function:

$$f : T'_u(t) \to \hat{y}$$

where $\hat{y}$ denotes the predicted class label for the trajectory. The model is evaluated based on its accuracy in correctly classifying trajectories according to this property.

## F.2  Experiment and Implementation Details

**Preprocessing**  We borrowed the experimental setup of AgentMove, similar to our zero-shot POI recommendation procedure in Section E.3. That is, we selected 200 random users from the test set and sampled one random trajectory for each user. This trajectory is then included in the test set. Each check-in is described by four attributes: the hour (in 12-hour format), the day of the week, the POI ID, and the POI category name.

LLMs are set to return outputs in a structured/JSON format, predicting whether the trajectory ended on a weekday or a weekend, along with an explanation of their reasoning. To ensure replicability, Gemini and GPT-4 models are set with the following generation parameters: a temperature of 0.0, a maximum output length of 1000 tokens, and an input context window capped at 2000 tokens. Due to API requirements, GPT-5 Nano uses the following generation parameters: a fixed temperature of 1.0, a maximum output length of 4096 tokens, low verbosity, and medium reasoning effort.

**Prompt**  Prompt template for spatiotemporal weekday-weekend classification is shown in Listing 4.

```
1 A trajectory is a sequence of check-ins, each represented as (start_time,
      poi_category). The detailed explanation of each element is as
      follows:
2 start_time: the start time of the check-in in 12h clock format.
3 poi_category: the category of the point of interest (POI) visited during
      the check-in
4
5 The trajectory is as follows: {[check-in time-of-day, POI category] for
      check-in in trajectory}
6
7 Your task is to classify whether the last check-in occurs on a weekday or
       a weekend.
8 Consider the temporal information (i.e., start_time) of the trajectory,
      which is important because people's activity varies during different
      time (e.g., nighttime versus daytime).
9 Consider the POI categories, which can provide insights into the user's
      activity patterns.
10 Also consider the city context, as different cities may have different
      cultural and social norms that influence activity patterns. The city
      is: {city}.
11
12 Please organize your answer in a JSON object containing following keys:
13 "prediction" ("weekday" or "weekend") and "reason" (a concise explanation
       that supports your prediction).
14 Do not include line breaks in your output.
```

Listing 4: Prompt for Weekday vs. Weekend Classification

**Models and Implementations**  We use the following LLMs in our experiments:

- Gemini 2.0 Flash (`gemini-2.0-flash`),

- GPT-4o Mini (`gpt-4o-mini`),
- GPT-4.1 Mini (`gpt-4.1-mini`),
- GPT-5 Nano (`gpt-5-nano`).

We accessed Gemini and GPT models via the official API. All modified code implementations are publicly available in our main dataset repository.

## G    LICENSE AND DATA USAGE

Our work **does not** involve the collection of new data. Instead, we derive our resulting dataset by combining and aligning two publicly available datasets, both of which are distributed under permissive licenses. We did not scrape data from the internet or use proprietary APIs to construct this dataset.

We accessed the Semantic Trails Dataset (Monti et al., 2018) via Figshare: `https://doi.org/10.6084/m9.figshare.7429076.v2`. The dataset is licensed under the Creative Commons CC0 1.0 license (`https://creativecommons.org/publicdomain/zero/1.0/`), which allows unrestricted copying, modification, and redistribution for any purpose, including commercial use, without requiring permission.

We accessed the Foursquare Open Source Places dataset via Hugging Face: `https://huggingface.co/datasets/foursquare/fsq-os-places`. Foursquare Open Source Places is licensed under the Apache License, Version 2.0:

> Copyright 2024 Foursquare Labs, Inc. All rights reserved.
> Licensed under the Apache License, Version 2.0 (the "License"); you may not use this file except in compliance with the License.
> You may obtain a copy of the License at: `http://www.apache.org/licenses/LICENSE-2.0`
> Unless required by applicable law or agreed to in writing, software distributed under the License is distributed on an "AS IS" BASIS, WITHOUT WARRANTIES OR CONDITIONS OF ANY KIND, either express or implied.
> See the License for the specific language governing permissions and limitations under the License.

More details are available in Foursquare's documentation: `https://docs.foursquare.com/data-products/docs/access-fsq-os-places`.

We will release our Massive-STEPS dataset under the same Apache Version 2.0 License, and have included Foursquare Open Source Places' license in our hosted dataset's README file.

## H    USAGE OF LLMS

While our experiments, particularly the zero-shot tasks, extensively studied LLM-based methods, we clarify that LLMs were not used in the preparation of this manuscript, except for minor grammatical corrections. All scientific content, analyses, and interpretations were developed solely by the authors.

