# OpenReview forum: "Massive-STEPS: Massive Semantic Trajectories for Understanding POI Check-ins"
_ICLR.cc/2026/Conference — Submitted to ICLR 2026_

### Official Review · Reviewer_Jv2g · 2025-10-26

**Soundness:** 3
**Presentation:** 3
**Contribution:** 2
**Rating:** 4
**Confidence:** 4

**Summary:**

The paper addresses the limitation of existing check-in datasets, which often lack coverage across diverse global regions. To overcome this issue, it introduces a new benchmark dataset called Massive-STEPS. This dataset encompasses data from 15 cities across a wide range of regions, including East, West, and Southeast Asia, North and South America, Australia, the Middle East, and Europe, covering time periods from 2012–2013 to 2017–2018. Moreover, it incorporates semantic attributes—such as POI names, categories, and addresses—that are useful for LLM-based methods. A variety of POI-related models are comprehensively evaluated on Massive-STEPS across multiple urban contexts.

**Strengths:**

1.	There is currently no widely accepted benchmark dataset, which leads to different models being trained and evaluated on different datasets, making fair comparison difficult. Therefore, developing a new public dataset with recent movement patterns, high diversity, and enriched semantics is highly meaningful for POI-related research.

2.	For POI modeling, existing datasets are often outdated, of low quality, lack regional diversity, or are not publicly available. The paper provides a thorough analysis of these limitations in existing POI datasets.

3.	The paper is clearly written and easy to follow, demonstrating good organization and readability.

**Weaknesses:**

1.	Although the dataset provides diverse data sources, it remains challenging for models to effectively learn from this diversity in Massive-STEPS. As shown in Table 3, each model is evaluated on a single sub-dataset of Massive-STEPS, resulting in no substantial difference in data usage compared to other datasets.

2.	Massive-STEPS is derived from the STD dataset with several preprocessing steps, such as trajectory grouping, city-level matching, and filtering. However, the transformation from STD to Massive-STEPS appears to be more of a data aggregation process rather than the construction of a fundamentally new dataset. Similar improvements in recency and diversity could potentially be achieved by aggregating existing POI datasets. Thus, the novelty of the dataset is somewhat limited.

3.	The paper would benefit from including additional descriptive statistics—such as the average trajectory length, mean sampling interval, or metrics that capture trajectory sparsity—to provide a clearer understanding of the dataset’s characteristics.

**Questions:**

1.	The authors claim that existing datasets suffer from low data quality. However, it is unclear how Massive-STEPS addresses or mitigates this issue. More details about the data cleaning or validation process would strengthen the paper.

2.	Does Massive-STEPS include or provide access to a road network map to support map-constrained methods or tasks? Clarifying this aspect would help readers understand the dataset’s applicability to spatially constrained modeling.

---

> ### Author Response · Authors · 2025-11-18
> **Response to Reviewer Jv2g (1/2)**
>
> > W1: Although the dataset provides diverse data sources, it remains challenging for models to effectively learn from this diversity in Massive-STEPS. As shown in Table 3, each model is evaluated on a single sub-dataset of Massive-STEPS, resulting in no substantial difference in data usage compared to other datasets.
>
> We would like to clarify that we do not intend to depart from the common practice of training on a single city (sub-dataset), as this approach remains widely used in the field. What distinguishes Massive-STEPS, as detailed in our Contributions section (L95-107), is its geographical diversity, high-quality check-ins, temporal relevance, longitudinal coverage, and completeness of POI information.
>
> Moreover, our dataset includes a substantially larger number of POIs (L309-318), which both (1) motivates the development of more computationally efficient models and (2) provides broader geographical breadth than existing POI datasets. These aspects are the core claims of our work and highlight how Massive-STEPS differs meaningfully from prior datasets.
>
> Finally, for researchers interested in training across multiple cities (e.g., for foundation model approaches), the dataset’s diverse city coverage readily supports such future studies.
>
> > W2: Massive-STEPS is derived from the STD dataset with several preprocessing steps, such as trajectory grouping, city-level matching, and filtering. However, the transformation from STD to Massive-STEPS appears to be more of a data aggregation process rather than the construction of a fundamentally new dataset. Similar improvements in recency and diversity could potentially be achieved by aggregating existing POI datasets. Thus, the novelty of the dataset is somewhat limited.
>
> We acknowledge the reviewer’s concern that our dataset may not be a fundamentally new dataset. A similar comment was raised by Reviewer mH1J (W1).
>
> However, we note that achieving “similar improvements in recency and diversity could potentially be achieved by aggregating existing POI datasets.” is currently *not* possible, to the best of our knowledge, as no publicly available datasets provide the necessary coverage and structure to support such an approach.
>
> Semantic Trails *is* the only closest alternative, but it lacks critical geographical coordinate information necessary for trajectory-based models (see Table 1). This limitation makes Semantic Trails unsuitable for a wide range of spatiotemporal prediction tasks and is a major reason for its underuse. Our work directly addresses this gap by aligning the dataset with FSQ-OS Places, enabling downstream trajectory modeling.
>
> Moreover, the urban and human mobility research community often relies on aggregated community or synthetic data (e.g., [1], [2], [3]) due to the absence of real-world, individual-level datasets. In contrast, our dataset provides real, individual-level trajectories, supporting more realistic and fine-grained mobility analysis. As noted in L36-39, synthetic or simulated data may not accurately reflect real-world human behavior, further highlighting the value of real-world data like ours. We therefore maintain that our dataset represents a substantial advancement in both usability and fidelity for spatiotemporal and human mobility research.
>
> > W3: The paper would benefit from including additional descriptive statistics—such as the average trajectory length, mean sampling interval, or metrics that capture trajectory sparsity—to provide a clearer understanding of the dataset’s characteristics.
>
> We thank the reviewer for this useful suggestion. We have updated Table 2 with additional descriptive statistics: average trajectory length (i.e., average number of check-ins per trajectory) and mean sampling interval (i.e., average time interval between check-ins). Please find them in the updated manuscript.

---

> > ### Author Response · Authors · 2025-11-18
> > **Response to Reviewer Jv2g (2/2)**
> >
> > > Q1: The authors claim that existing datasets suffer from low data quality. However, it is unclear how Massive-STEPS addresses or mitigates this issue. More details about the data cleaning or validation process would strengthen the paper.
> >
> > As detailed in Section 3.1, the quality of Massive-STEPS is ensured by inheriting Semantic Trails’ (STD) strict filtering procedures (L195-197), which already remove low-quality, erroneous entries that affect prior existing datasets (e.g., GSCD, see L190-205).
> >
> > Because of this already-strict data filtering conducted by STD, our additional processing focused only on removing inactive users and short trajectories (L284-286), a standard practice in trajectory datasets. Our resulting Tokyo subset, for example, is smaller than the existing FSQ-TKY counterpart, reflecting the effectiveness of STD’s prior filtering to remove erroneous entries (L304-306). Therefore, we believe no further cleaning was necessary, as the inherited filtering already ensures high data quality while preserving meaningful user trajectories.
> >
> > > Q2: Does Massive-STEPS include or provide access to a road network map to support map-constrained methods or tasks? Clarifying this aspect would help readers understand the dataset’s applicability to spatially constrained modeling.
> >
> > We thank the reviewer for the suggestion. While providing a road network map could be useful for some spatially constrained modeling tasks, it is outside the scope of our dataset and benchmark, and was not claimed as a contribution. That said, researchers can readily integrate such a map if desired. For example, a road network can be obtained from OpenStreetMap, and since we release all geographical boundaries (GeoJSON files) for each city, any extracted network can be precisely aligned with the regions used in Massive-STEPS. We view this as a natural extension for future work and a capability that our dataset facilitates.
> >
> > [1] Nilforoshan, Hamed, et al. "Human mobility networks reveal increased segregation in large cities." Nature 624.7992 (2023): 586-592.
> >
> > [2] Jiang, Yuqin, et al. "Comparative analysis of human mobility patterns: utilizing taxi and mobile (SafeGraph) data to investigate neighbourhood-scale mobility in New York City." Annals of GIS (2025): 1-25.
> >
> > [3] Stanford, Chris, et al. "Numosim: A synthetic mobility dataset with anomaly detection benchmarks." Proceedings of the 1st ACM SIGSPATIAL International Workshop on Geospatial Anomaly Detection. 2024.

---

### Official Review · Reviewer_EkXJ · 2025-10-28

**Soundness:** 3
**Presentation:** 3
**Contribution:** 3
**Rating:** 6
**Confidence:** 5

**Summary:**

This paper introduces Massive-STEPS, a large new benchmark dataset for POI trajectory modeling. Its purpose is to address critical limitations in current human mobility research, which heavily relies on outdated datasets from 2012–2013 and is disproportionately focused on a handful of cities like New York and Tokyo. Furthermore, many existing datasets suffer from poor data quality (e.g., GSCD contains nearly 50% erroneous entries) and lack reproducibility. To address these issues, Massive-STEPS is built upon the high-quality STD dataset, providing data across 15 geographically and culturally diverse cities, including understudied regions. The dataset incorporates both recent (2017–2018) and earlier (2012–2013) check-in data, spanning a total duration of 24 months. It is further semantically enriched using Foursquare OS Places data, supplementing metadata such as POI coordinates, names, and addresses. Finally, the authors conduct extensive benchmarking on Massive-STEPS across three tasks, supervised POI recommendation, zero-shot POI recommendation, and spatio-temporal classification, to advance reproducible and equitable mobility research.

**Strengths:**

1. The first strength of this paper lies in the dataset itself. It addresses a widely recognized, severe bottleneck that has hindered progress in the field. By providing a large-scale, more modern, geographically diverse, and reproducible dataset, this work offers an invaluable service to the community.
2. The author conducted extensive experiments covering a wide range of approaches, from classical models, GNNs, and LLMs. Also, they were tested across three distinct tasks, which provides a highly robust and valuable baseline for future research utilizing this dataset.
3. The dataset provided encompasses 15 distinct cities, particularly including some less popular or previously overlooked regions, marking a significant advancement. Combined with data from 2017-2018, this facilitates more generalized and practically relevant analysis of human mobility.

**Weaknesses:**

1. Although this paper provides benchmarking and implementation code, consolidating all models used into a standardized code repository would significantly enhance the quality of the work.
2. This paper primarily reviews and integrates existing work, without proposing any independent models or insights.
3. It would be better if the authors compared the performance of the model in terms of changes in the dataset between the 2012-2013 and 2017-2018 time periods.

**Questions:**

Please refer to the weaknesses

---

> ### Author Response · Authors · 2025-11-18
> **Response to Reviewer EkXJ**
>
> > W1: Although this paper provides benchmarking and implementation code, consolidating all models used into a standardized code repository would significantly enhance the quality of the work.
>
> We thank the reviewer for the suggestion. Like the reviewer mentioned, currently, all models used in our benchmarks are publicly available under submodules, which ensures that each model retains its original structure and dependencies.
>
> We understand the reviewer’s suggestion to consolidate all models into a single unified repository. While this could simplify the codebase, it would require modifying existing codebases, which may introduce inconsistencies or potential reproducibility issues. We have ensured that our current structure allows users to clone the repository and run all benchmarks end-to-end, and we provide detailed instructions for running each model.
>
> We are open to future work on further unifying the repository if community demand arises, but we believe the current setup already supports fully reproducible benchmarking without compromising correctness or transparency.
>
> > W2: This paper primarily reviews and integrates existing work, without proposing any independent models or insights.
>
> We appreciate the reviewer’s perspective and would like to clarify the intended contribution of our work. While it is correct that we do not introduce a new model, this is not the primary goal of our paper. As indicated in the submission, our primary area of contribution lies in “datasets and benchmarks”. Massive-STEPS provides a family of datasets along with benchmark evaluations of 10 POI recommendation models (supervised and zero-shot) and 4 LLMs for spatio-temporal classification, totaling over 300 experiments. While we do not introduce new models/algorithms, our dataset *enables* future methodological innovations by providing the first large-scale, multi-city resource of this kind.
>
> Importantly, existing “foundation model” approaches claiming multi-city adaptability do not perform well under these conditions (see L371-373), highlighting that scaling across cities is non-trivial as previously claimed. Massive-STEPS enables new insights like these that were previously impossible, including but not limited to:
>
> 1. Evaluating whether models truly generalize across diverse and underrepresented cities, identifying which models train effectively and which fail despite foundation model claims.
> 2. Formulating a strong hypothesis relating a city’s category entropy to model performance, which previous studies could not test due to limited city coverage (L374-377).
> 3. Demonstrating that zero-shot methods (Task 2) can be competitive with supervised approaches (Task 1) across different cities.
> 4. Showing that spatiotemporal and semantic mobility patterns vary across cities, impacting downstream tasks such as spatio-temporal classification (Task 3).
>
> In short, although we do not introduce independent models, our *dataset and benchmark directly support new insights* into model behavior, cross-city generalization, and mobility patterns, which constitute a significant contribution of this work.
>
> > W3: It would be better if the authors compared the performance of the model in terms of changes in the dataset between the 2012-2013 and 2017-2018 time periods.
>
> We sincerely thank the reviewer for this insightful suggestion! In response, we conducted new experiments and additional analyses comparing model performance across the 2012-2013 and 2017-2018 time periods.
>
> That is, we split the benchmark results for Task 2 by time period and evaluated multiple LLMs using the LLM-Move prompting method, which we identified as the strongest-performing approach (Table 4; L416-420). This allowed us to uncover several new insights:
>
> 1. In nearly all cities, except Jakarta and Tangerang, LLM accuracy decreased in the 2017-2018 period. The most pronounced decline in performance occurred in Kuwait City. This indicates that user trajectories and behaviors in later years tend to be more challenging to predict than those from 2012-2013.
> 2. Across the LLMs evaluated, Gemini 2 Flash remained the overall best-performing model across both time periods, further highlighting its capabilities for zero-shot next POI recommendation tasks irrespective of time periods.
>
> We have added these new results to our manuscript (results in Table 5, details on L437-454).

---

### Official Review · Reviewer_mH1J · 2025-10-30

**Soundness:** 2
**Presentation:** 1
**Contribution:** 1
**Rating:** 2
**Confidence:** 4

**Summary:**

This paper presents Massive-STEPS, a large-scale dataset designed to address longstanding limitations in POI trajectory modeling. Specifically, the reliance on outdated, geographically limited, and non-reproducible check-in datasets. Massive-STEPS provides a semantically enriched resource covering 15 cities across diverse global regions and two time periods, enabling both longitudinal and cross-city analyses. The dataset includes rich semantic information such as venue names, addresses, categories, and coordinates. The authors further provide benchmark results for both supervised and zero-shot POI trajectory modeling methods, demonstrating the dataset’s potential utility across different model types and tasks.

**Strengths:**

1. The paper provides a solid overview of existing check-in datasets and clearly identifies their limitations, offering useful context for the community.

**Weaknesses:**

1. The contribution appears somewhat incremental. As shown in Table 1, Massive-STEPS seems closely related to Semantic Trails, differing mainly through data reorganization rather than introducing substantial new content or methodology.

2. The authors emphasize that prior datasets are outdated; however, Massive-STEPS itself relies on data from 2017–2018, which remains quite old by 2025 standards and does not fully address the claimed issue of temporal relevance.

3. Although the paper defines three different benchmark tasks, the corresponding model evaluations are limited in both scale and complexity, lacking deeper exploration or meaningful analysis.

4. The writing quality could be improved. While the paper is lengthy, the information density is relatively low—many pages are dominated by large tables or figures containing simple statistics (e.g., Figure 1 visualizes only 14 data points yet occupies a full page).

**Questions:**

Could the authors elaborate on the weaknesses.

---

> ### Author Response · Authors · 2025-11-18
> **Response to Reviewer mH1J (1/2)**
>
> > W1: The contribution appears somewhat incremental. As shown in Table 1, Massive-STEPS seems closely related to Semantic Trails, differing mainly through data reorganization rather than introducing substantial new content or methodology.
>
> We appreciate the reviewer’s perspective and would like to clarify why we believe the contribution extends beyond an incremental update.
>
> First, as shown in Table 1, Semantic Trails lacks critical *geographical coordinate* information (latitude and longitude) required for trajectory-based models. This omission makes Semantic Trails unusable for a wide range of spatiotemporal prediction tasks, hence why it is underused. Our work directly addresses this limitation by aligning the dataset with FSQ-OS Places, thereby filling a key gap and enabling downstream trajectory modeling.
>
> Second, one of the core contributions of our work is not merely data reorganization but *benchmark standardization*, a pressing need in the POI recommendation and prediction field (not to mention our contribution of extensive benchmarking and experiments across a diverse range of cities). As discussed in L207-215 and summarized in Table 6, most existing studies are *not reproducible* unless they rely on FSQ-NYC/TKY. Our dataset and benchmarks resolve this by being fully transparent and open-source, from data construction to benchmark evaluation. Without such a standardized foundation, fair comparative analysis remains impossible, as each study operates under distinct settings. Similar to how standardized datasets have advanced fields like computer vision and NLP, we believe that establishing such a benchmark is essential for progress in this domain.
>
> Finally, while we do not introduce a new methodology (e.g., a model or algorithm), our dataset *enables* such future methodological innovations. It also broadens research opportunities for the human mobility community, which has relied on aggregated or synthetic data (e.g., [1], [2], [3]). In contrast, our dataset offers real, individual-level trajectories, providing a foundation for more realistic and fine-grained mobility analysis.
>
> Considering our contributions and the literature gaps identified in our manuscript, we believe that our work makes a substantial contribution to the dataset and benchmarking landscape in human mobility research.
>
> > W2: The authors emphasize that prior datasets are outdated; however, Massive-STEPS itself relies on data from 2017–2018, which remains quite old by 2025 standards and does not fully address the claimed issue of temporal relevance.
>
> We appreciate the reviewer's attention to the temporal scope of our dataset.
>
> Firstly, we acknowledge that Massive-STEPS, which includes data from 2017-2018, does not capture the most current urban mobility patterns of 2025. We agree there is a clear and continuing need for newer datasets covering the post-2020 period.
>
> Secondly, we want to clarify that Massive-STEPS was not presented as a reflection of *present-day* mobility dynamics (L505-507). Rather, our primary motivation was to address the continued, widespread reliance on older datasets such as FSQ-NYC and TKY, which date back to 2012-2013 and are still commonly used (L156-157). Massive-STEPS effectively bridges this temporal gap, offering a more recent baseline for future research.
>
> Thirdly, we want to clarify that we did not claim that these older datasets are *outdated* and imply that they are no longer valuable. In fact, we intentionally included the 2012-2013 timespan within Massive-STEPS (L99-102) precisely because this inclusion enables longitudinal studies, a crucial research dimension that prior datasets cannot support.
>
> We believe that, while not perfectly contemporary, Massive-STEPS offers a meaningful step forward by being a substantially larger, more reproducible, and more recent alternative to the prevalent 2012-2013 benchmarks. To the best of our knowledge, there are no alternative, publicly available datasets that provide the same or newer temporal timespan as our dataset.

---

> ### Author Response · Authors · 2025-11-18
> **Response to Reviewer mH1J (2/2)**
>
> > W3: Although the paper defines three different benchmark tasks, the corresponding model evaluations are limited in both scale and complexity, lacking deeper exploration or meaningful analysis.
>
> We appreciate the reviewer’s feedback and would like to clarify that our model evaluations are, in fact, extensive in both scale and methodological diversity. In Tasks 1 and 2, we benchmarked *10 baseline methods across 15 cities*, encompassing a diverse range of models: from classical Markov chains and deep learning architectures to state-of-the-art GNNs and even modern LLMs, both open- and closed-source. Altogether, this involved *over 250 experiments* (105 supervised and 180 zero-shot), ensuring both robustness and comprehensive coverage. We believe this substantially exceeds the scale and diversity of most prior POI prediction studies, many of which report results on only 2-3 cities.
>
> As shown in Table 6, while newer models exist, many are not reproducible or publicly available, making them unsuitable as baselines. We intentionally focused on open-source and reproducible methods to ensure fairness and transparency in evaluation. The resulting analyses (Tables 3-6) already yield meaningful insights into how model performance varies across cities and task formulations, thereby providing a valuable understanding of model behavior and dataset characteristics.
>
> > W4: The writing quality could be improved. While the paper is lengthy, the information density is relatively low—many pages are dominated by large tables or figures containing simple statistics (e.g., Figure 1 visualizes only 14 data points yet occupies a full page)
>
> We appreciate the reviewer’s feedback and would like to request clarification regarding the concerns about writing quality and information density.
>
> First, it appears there may be a misunderstanding about Figure 1. The figure does not visualize 14 data points; rather, it provides a high-level schematic of all benchmark tasks included in Massive-STEPS. Moreover, Figure 1 occupies only about ten lines on that page, while the remaining lines contain extensive text describing our dataset, benchmark tasks, and overall contributions. *None of our figures or tables occupies a full page*. That said, we understand the concern about space and would be open to reconsidering the placement of Figure 1 in the appendix if necessary. However, we consider it essential to the main text because it serves as a visual abstract summarizing our benchmark at a glance.
>
> Second, we would like to emphasize that each figure and table in the main paper presents essential information about the dataset and benchmark results. Within the 9-page constraint, we have carefully balanced clarity, completeness, and conciseness. We thoroughly covered the existing literature gap, the dataset construction process, benchmarks, and empirical analysis. Additional implementation and analysis details are even provided in the appendix due to space limitations and the level of detail we have to include in our work.
>
> Given this context, we kindly ask the reviewer to reconsider the comment and presentation score, and we would be grateful for any specific suggestions on figures that may appear unclear or overly large.
>
> [1] Nilforoshan, Hamed, et al. "Human mobility networks reveal increased segregation in large cities." Nature 624.7992 (2023): 586-592.
>
> [2] Jiang, Yuqin, et al. "Comparative analysis of human mobility patterns: utilizing taxi and mobile (SafeGraph) data to investigate neighbourhood-scale mobility in New York City." Annals of GIS (2025): 1-25.
>
> [3] Stanford, Chris, et al. "Numosim: A synthetic mobility dataset with anomaly detection benchmarks." Proceedings of the 1st ACM SIGSPATIAL International Workshop on Geospatial Anomaly Detection. 2024.

---

> > ### Comment · Reviewer_mH1J · 2025-11-24
> >
> > Thank you for your response. First, I apologize that my wording in W4 was unclear. What I meant is that the information content of Figure 2 is relatively low and is too large because it contains only 14 points. Squeezing it to one column or unfolding these points might present the results more effectively.
> >
> > Regarding W1, I appreciate your effort in standardizing the dataset, but I believe this contribution alone is not sufficient to support a conference of ICLR’s caliber.
> >
> > Regarding W2, I still feel that the difference between 2017 and 2013 is not very perceptible, at least from my perspective, and both time periods are before the COVID-19 pandemic. Therefore, their relevance for understanding modern human mobility patterns is limited.
> >
> > Regarding W3, I still think the results in Table 4 are somewhat thin and simplistic. It looks as though the authors tried to cover every aspect, but none of them in enough depth.

---

> ### Author Response · Authors · 2025-11-25
> **Response to Official Comment by Reviewer mH1J (1/3)**
>
> We thank the reviewer for their continued engagement. We address the follow-up concerns below.
>
> > What I meant is that the information content of Figure 2 is relatively low and is too large because it contains only 14 points. Squeezing it to one column or unfolding these points might present the results more effectively.
>
> We accept the recommendation to improve the space efficiency of Figure 2 and will resize it to a single column or condense the visualization in the final revision. We are also willing to move it to the appendix to prioritize additional analysis if needed.
>
> We wish to clarify that the figure's intent was to visually demonstrate the global geographical coverage of Massive-STEPS. As we argued throughout the paper, prior work is often limited to just 2-3 cities, or heavily focused on two particular cities, NYC and Tokyo. Visualizing our coverage across 15 cities and multiple continents serves as immediate evidence of how our work breaks this status quo.
>
> Crucially, we ask the reviewer to reconsider the given presentation score. A score of 1 typically implies that a paper is unreadable, disorganized, or structurally flawed. Penalizing the entire manuscript’s presentation based on the layout of a single figure seems disproportionate, particularly given that the other reviewers rated the presentation as 3 (good) and found the text clear and the technical depth sufficient. We believe the manuscript’s overall clarity warrants a score that reflects the quality of the writing and organization, rather than an issue with one minor visualization.
>
> > Regarding W1, I appreciate your effort in standardizing the dataset, but I believe this contribution alone is not sufficient to support a conference of ICLR’s caliber.
>
> We strongly maintain that Massive-STEPS fills a critical infrastructure gap that currently stifles the human/urban mobility community. We urge the reviewer to evaluate our contribution through the lens of the specific challenges facing this domain, rather than the standards of more mature fields like CV or NLP, for example.
>
> First, as summarized in Table 7, the field of human mobility modeling lags significantly behind other domains precisely because it lacks the foundational infrastructure that Massive-STEPS provides. While CV and NLP enjoy standardized benchmarks that allow for focus on methodological nuance, mobility research is currently fragmented between outdated datasets and inaccessible proprietary data. Consequently, claims of “state-of-the-art” performance in our field are often unverifiable. For instance, the authors of STHGCN (in their official repository) noted the difficulty of fairly comparing their work against strong baselines like GETNext due to different experimental settings, while recent works like UniMove rely on proprietary data with no description of sourcing or construction. By introducing a fully replicable, open-source benchmark across 15 cities, our work is not merely “reorganizing” data; it is resolving a systemic reproducibility issue that prevents definitive measurement of progress.
>
> Second, we posit that in the context of dataset & benchmark tracks at top-tier venues like ICLR (to which we are submitting), value is derived from scale, utility, and standardization, not necessarily the introduction of novel model architectures (as the reviewer remarked in W1). Top-tier conferences consistently accept papers that build the infrastructure required to train and evaluate existing models, without necessarily proposing a new model/method. For example, ConvCodeWorld [1] (Han et al., ICLR 2025) standardizes code-generation benchmarks to improve replicability; STARK [2] (Quan et al., NeurIPS 2025) offers a benchmark for spatiotemporal reasoning in LLMs; and K-HALU [3] (Seo and Lim, ICLR 2025) transforms an existing Korean dataset into synthetic hallucinations to evaluate LLM performance. Similarly, Massive-STEPS provides the first standardized evaluation protocol and the largest open-source benchmark for POI trajectories. This aligns perfectly with the caliber of top-tier contributions that “unblock” research areas where data and benchmarking fragmentation have made rigorous evaluations difficult.
>
> Third, regarding the characterization of our work as “incremental” (W1), we believe this overlooks the distinction between incremental improvement and foundational standardization. As noted in our initial response, datasets like Semantic Trails are currently functionally unusable for spatial representation learning due to the lack of coordinates. By fusing this data with FSQ-OS and standardizing it into a valid training resource, we are effectively “resurrecting” unusable data. Therefore, our contribution is not just an update; it is the establishment of a standardized “playing field” that allows future methodological innovations to be fairly compared and reproduced; a standard that does not currently exist in human mobility research.

---

> ### Author Response · Authors · 2025-11-25
> **Response to Official Comment by Reviewer mH1J (2/3)**
>
> > Regarding W2, I still feel that the difference between 2017 and 2013 is not very perceptible, at least from my perspective, and both time periods are before the COVID-19 pandemic. Therefore, their relevance for understanding modern human mobility patterns is limited.
>
> We would like to show that the difference between 2017 and 2013 mobility data **is** very much substantial and meaningful.
>
> First, we added a new figure (Fig. 6) illustrating the temporal shift in the distribution of the top-10 most visited POI categories across the two periods. This reveals clear distributional changes, showing that visitation behaviors can shift substantially *even* within the same city. Such analyses were previously not feasible due to the limited temporal scope of earlier datasets.
>
> Second, to further establish the relevance of the 2017-2018 data (and the longitudinal feature of our dataset), we conducted an additional analysis of POI-level turnover by calculating how many POIs ever opened, had closed by 2025, and had closed specifically within 2014-2016, the temporal gap in our dataset. The following table shows our findings:
>
> | City | POIs Ever Opened | Total POIs Confirmed Closed (up to 2025) | Closed within 2014-2016 |
> |---|--:|--:|--:|
> | New York | 49218 | 13,009 (26.43%) | 3,118 (6.34%) |
> | Melbourne | 7699 | 1,850 (24.03%) | 209 (2.71%) |
> | Sydney | 8986 | 1,759 (19.57%) | 253 (2.82%) |
> | Moscow | 17822 | 3,021 (16.95%) | 868 (4.87%) |
> | São Paulo | 38377 | 4,990 (13.00%) | 1,257 (3.28%) |
> | Shanghai | 4462 | 661 (14.81%) | 81 (1.82%) |
> | Tokyo | 4725 | 421 (8.91%) | 0 (0.00%) |
> | Petaling Jaya | 60158 | 4,186 (6.96%) | 1,533 (2.55%) |
> | Istanbul | 53812 | 2,833 (5.26%) | 481 (0.89%) |
> | Beijing | 1127 | 56 (4.97%) | 10 (0.89%) |
> | Jakarta | 76116 | 3,527 (4.63%) | 483 (0.63%) |
> | Bandung | 29026 | 1,053 (3.63%) | 182 (0.63%) |
> | Palembang | 4343 | 143 (3.29%) | 23 (0.53%) |
> | Tangerang | 12956 | 383 (2.96%) | 50 (0.39%) |
> | Kuwait City | 17180 | 161 (0.94%) | 22 (0.13%) |
>
> This analysis reveals a highly volatile POI landscape, with substantial closure rates in major cities such as New York (26.43%), Melbourne (24.03%), and Sydney (19.57%), demonstrating that the POI environment is far from static and underscoring the need for a dataset that spans multiple time periods. The absolute number of POIs that closed during the 2014-2016 gap further reinforces this point: even within this narrow three-year window, more than 3,000 POIs closed in New York and over 1,000 closed in São Paulo and Petaling Jaya, confirming that significant structural changes occurred between our two observation points.
>
> Third, following Reviewer EkXJ’s suggestion, we conducted new experiments comparing LLM performance across the two time periods. These results, now included in Table 5 and L420-427, show that model performance is not static: it varies noticeably depending on both the city and the timespan under consideration. This reinforces one of the central contributions of our work: Massive-STEPS enables longitudinal evaluation and reveals that 2017-2018 data provides distinct and valuable insights.
>
> | Time Period | Model | Bandung | Beijing | Istanbul | Jakarta | KC | Melbourne | Moscow | NY | Palembang | PJ | SP | Shanghai | Sydney | Tangerang | Tokyo |
> |---|---|:---:|:---:|:---:|:---:|:---:|:---:|:---:|:---:|:---:|:---:|:---:|:---:|:---:|:---:|:---:|
> | 2012-2013 | Gemini 2 Flash  | 0.227 | 0.102 | 0.212 | 0.295 | 0.423 | 0.226 | 0.218 | 0.240 | 0.256 | 0.199 | 0.298 | 0.192 | 0.256 | 0.197 | N/A |
> |  | Qwen 2.5 7B | 0.098 | 0.204 | 0.192 | 0.114 | 0.269 | 0.116 | 0.234 | 0.130 | 0.128 | 0.142 | 0.173 | 0.109 | 0.122 | 0.172 | N/A |
> |  | Llama 3.1 8B | 0.031 | 0.041 | 0.007 | 0.010 | 0.000 | 0.039 | 0.005 | 0.032 | 0.010 | 0.014 | 0.048 | 0.006 | 0.064 | 0.000 | N/A |
> |  | Gemma 2 9B  | 0.180 | 0.102 | 0.116 | 0.228 | 0.308 | 0.097 | 0.112 | 0.130 | 0.215 | 0.199 | 0.202 | 0.109 | 0.122 | 0.126 | N/A |
> | 2017-2018 | Gemini 2 Flash  | 0.167 | 0.000 | 0.185 | 0.286 | 0.190 | 0.222 | 0.333 | 0.217 | 0.400 | 0.237 | 0.219 | 0.091 | 0.136 | 0.500 | 0.250 |
> |  | Qwen 2.5 7B | 0.167 | 0.000 | 0.130 | 0.143 | 0.144 | 0.089 | 0.000 | 0.087 | 0.200 | 0.119 | 0.063 | 0.045 | 0.136 | 0.500 | 0.250 |
> |  | Llama 3.1 8B | 0.000 | 0.333 | 0.037 | 0.143 | 0.011 | 0.044 | 0.000 | 0.043 | 0.000 | 0.102 | 0.031 | 0.068 | 0.023 | 0.000 | 0.030 |
> |  | Gemma 2 9B  | 0.000 | 0.000 | 0.056 | 0.429 | 0.092 | 0.178 | 0.000 | 0.065 | 0.000 | 0.119 | 0.156 | 0.091 | 0.136 | 0.000 | 0.130 |

---

> > ### Author Response · Authors · 2025-11-25
> > **Response to Official Comment by Reviewer mH1J (3/3)**
> >
> > Finally, while we agree that post-COVID mobility data would be highly desirable, no publicly available dataset of this scale and detail exists, largely due to privacy restrictions. The research community’s options are currently limited to using the 2012-2013 datasets **or** upgrading to the substantially more recent, diverse, and reproducible 2017-2018 Massive-STEPS. Rejecting the latter for not meeting the ideal (but *currently* impossible) standard of containing post-COVID data effectively penalizes the field and halts the adoption of a much-needed, updated infrastructure.
> >
> > > Regarding W3, I still think the results in Table 4 are somewhat thin and simplistic. It looks as though the authors tried to cover every aspect, but none of them in enough depth.
> >
> > We understand the reviewer’s desire for deeper analysis; however, this expectation does not align with the goal of our benchmark paper, which is to establish breadth and robust baselines. Our model evaluation is intentionally designed to prioritize broad coverage over narrow depth, particularly given the limited geographical scope of prior studies. We demonstrate performance across 15 cities (a first in this domain), three distinct tasks, and 14 model baselines (including DL architectures and LLMs), evaluated under supervised and zero-shot settings.
> >
> > For example, the Table 4 under critique is the first to benchmark four LLMs using three prompting methods across the global geographical diversity of 15 cities. This scope provides a reliable performance benchmark for the community, following the precedent of related spatio-temporal benchmarks such as STARK (Quan et al., NeurIPS 2025), which also focuses on breadth. Conducting an in-depth diagnosis of a single model’s failure modes, for example, would have constrained the paper’s utility as a general benchmark.
> >
> > We further note that our analysis of Task 2 results is not limited to Acc@1 metrics, which alone could be misleading. Additional metrics, such as NDCG for reranking, have been reported in Appendix E.4 (spread across three tables due to space constraints). Overall, we believe that our broad, rigorous, and novel benchmarking substantially exceeds the standard for initial dataset releases and is far from “thin.”
> >
> > [1] Quan, P., Wang, B., Yang, K., Han, L., & Srivastava, M. (2025). Benchmarking spatiotemporal reasoning in LLMs and reasoning models: Capabilities and challenges.
> >
> > [2] Han, H., Hwang, S.-W., Samdani, R., & He, Y. (2025). ConvCodeWorld: Benchmarking conversational code generation in reproducible feedback environments.
> >
> > [3] Seo, J., & Lim, H. (2025). K-HALU: Multiple Answer Korean Hallucination Benchmark for Large Language Models.

---

### Author Response · Authors · 2025-12-01
**Consolidated Response Summary for Reviewers**

Dear Reviewers,

We thank you for your reviews and discussion, which allows us to improve our manuscript and work. We have taken all your feedback, made clarifications, and revised our manuscript accordingly. For each weakness or question raised, we restate your point, provide our response, and indicate the corresponding revisions made in the manuscript. We have uploaded our latest manuscript with these changes and highlighted additions in blue for your convenience.

Below, we provide a consolidated response to points raised by multiple reviewers to streamline the rebuttal process.

- **Contribution and Novelty**: We clarified that our work is not merely data aggregation but a foundational standardization necessary for the field of human and urban mobility to progress in terms of reproducibility, as we’ve extensively detailed dataset reproducibility issues in recent human mobility studies (Table 7, Appendix A). This aligns our contribution with the need for systemic reproducibility in the field, not a new and better model (L96-126).

- **Temporal Relevance and Recency & Longitudinal Coverage**: While our data is pre-COVID, we demonstrated that Massive-STEPS offers a more recent and diverse alternative to the prevalent 2012-2013 benchmarks. Crucially, we:

    1. Used new analysis of POI-level turnover, with over 24% closures in major cities, showing that the POI landscape is non-static and highly volatile (see Table 9, Appendix B),
    2. Showed substantial user behavior distribution shifts across the two time spans, despite the perceived short 5-year gap (see L33-338, Fig. 6, Appendix B), and
    3. Conducted additional longitudinal experiments comparing 2012-2013 vs. 2017-2018 LLM performance (see L452-459, Table 5).

    These confirmed substantial differences and distinct, more challenging user behaviors in the later period, validating our dataset's unique temporal and longitudinal value.

- **Evaluation Breadth**: We clarify that our evaluation's primary goal is breadth for benchmarking across 15 cities, 3 tasks, 7 DL models, and 7 LLMs, which is a first-of-its-kind scale in this domain. This comprehensive scope, involving over 300+ experiments, yields key insights (e.g., zero-shot competitiveness, foundation model limitations) that would have been missed by focusing on deep, narrow analysis of a single model, thereby fulfilling the requirements of a strong benchmark paper.

---

### Author Response · Authors · 2025-12-01
**Summary for AC and Reviewers**

Dear ACs and Reviewers,

We thank your effort in reviewing our work and the discussion. This comment summarizes the key points discussed during the rebuttal period for our submission, Massive-STEPS.

Reviewer **mH1J**: "The paper provides a solid overview of existing check-in datasets and clearly identifies their limitations, offering useful context for the community."

Reviewer **EkXJ**: "The first strength of this paper lies in the dataset itself. It addresses a widely recognized, severe bottleneck that has hindered progress in the field."

Reviewer **Jv2g**: There is currently no widely accepted benchmark dataset, which leads to different models being trained and evaluated on different datasets, making fair comparison difficult. Therefore, developing a new public dataset with recent movement patterns, high diversity, and enriched semantics is highly meaningful for POI-related research."

---

### Addressing Reviewer Concerns

There were initial concerns from the reviewers, and thanks to their feedback, we have added necessary clarification, conducted additional experiments and analysis, and have integrated them into our revised manuscript.

Key Features of Massive-STEPS

- **Standardized, Extensible, Diverse**: We highlighted that Massive-STEPS fills a critical dataset gap in the human/urban mobility community, which lacks standardized, reproducible benchmarks, in contrast to more mature domains like CV/NLP. Our work leverages and “resurrects” Semantic Trails by fusing it with FSQ OS Places and standardizing it into a usable, reproducible benchmark across 15 diverse cities and 3 benchmark tasks. Massive-STEPS offers an alternative to aggregated and simulated data, providing real-world, individual trajectories that are of higher fidelity (L35-40).

- **Temporally More Recent and Longitudinal Coverage**: Although our data is limited to 2017-2018 and is not post-2020, our dataset offers a more recent, diverse, and reproducible alternative to the prevalent 2012-2013 benchmarks. This is currently our limitation, and we’d like to very much integrate post-COVID data once it is publicly available (L509-511). Still, we demonstrated the value and uniqueness of our dataset's temporal coverage across 2 time spans (see details below).

- **Benchmark Breadth**: As a benchmark paper, our evaluation is intentionally designed for breadth. We provide the first-ever benchmark of this scale (7 DL models, 7 LLMs, 15 cities, 3 tasks, supervised & zero-shot settings). We have performed over 300+ experiments encompassing a wide range of models. This breadth directly supports new insights, such as demonstrating that zero-shot methods can be competitive with supervised approaches and identifying that cross-city scaling is non-trivial despite "foundation model" claims.

Importance of Longitudinal Datasets, Analyses, and Studies

Our dataset also covers two time spans: 2012-2013 and 2017-2018. Despite the seemingly short 5-year gap, we show the following:

- **POI Closures**: Thanks to Reviewer **mH1J**, we conducted an additional analysis of POI-level turnover by calculating closure rates, with data from FSQ OS Places. For major cities like New York and Melbourne, over 24% of POIs had closed by 2025, demonstrating the highly volatile and non-static environment (L322-323).

- **Model Performance Across Timespans**: Thanks to Reviewer **EkXJ**, we analyzed LLM performance across the two periods, and showed that model performance is not static (Table 5). LLM accuracy generally decreased in the later period, indicating that 2017-2018 user behaviors are distinctly more challenging to predict. (L452-459).

- **User Behavior Distribution Shifts**: We added a new figure (Fig. 6) showing the temporal shift in the distribution of the top-10 most visited POI categories, revealing clear distributional changes even across the perceived short 5-year gap (L336-337).

We again thank the reviewers for their helpful feedback and their recognition that Massive-STEPS fills a critical gap in the domain of human and urban mobility. We believe that our revised manuscript further highlights the necessity and significance of our dataset, thanks to their feedback.

---

### Meta-Review · Area_Chair_1jmm · 2025-12-25

**Summary:**

This paper presents an extension of the Semantic Trails datasets. These datasets contain metadata of checkins of users at points of interest (POIs), obtained from Foursquare. The POIs are venues such as landmarks, restaurants, and shops. The added value of the original dataset has been enriching the data with additional semantic information such as GeoNames / Wikidata and hierarchical categorization. The present paper builds upon the original dataset and argues that current research suffers from two limitations. 1) It is using old data (largely from 2012 - 2013), and 2) It does not reflect geographic diversity. The new dataset uses more recent data covering a longer duration and spans 15 geographically diverse cities. The ultimate goal this extension enables is more equitable research.

**Reviewer Concerns:**

The main criticism by the reviewers is the limited novelty of the work because it does not present new data and is seen as mainly a transformation of STD and additional benchmarks.  The reviewers also mention more technical criticisms such as limited evaluations of defined benchmark tasks (mH1J), lack of performance comparisons between the different periods (EkXJ), and the need for more descriptive statistics (Jv2g).

**Reviewer Scores:**

The authors answered some of the more technical questions to a level that would have likely addressed some of the reviewers' concerns. As for the novelty, the matter is more difficult to judge. The authors rightly made the argument that for a dataset and benchmark contribution, a novel method performing well on a given task is not required. They do not sufficiently address the reviewers' other core concern about the fact that the work claims to propose a new dataset, while what it essentially delivers is a reorganization of already publicly available data, along with providing a benchmark. The authors rightly point out that this has some value to the community, and this is, in part, recognized by the reviewers. It remains doubtful that this argument will be sufficient for the more critical reviewers to increase the score because ultimately, the claim of the work of providing a new dataset seems not to match the reviewers' expectations of what constitutes such a dataset and its usefulness to the community.

---

### Decision · Program_Chairs · 2026-01-26

Reject